# Learning and Evaluating Representations for Deep One-class Classification

**Kihyuk Sohn**,* **Chun-Liang Li**\*, **Jinsung Yoon, Minho Jin & Tomas Pfister**
Google Cloud AI
{kihyuks,chunliang,jinsungyoon,minhojin,tpfister}@google.com

## Abstract

We present a two-stage framework for deep one-class classification. We first learn self-supervised representations from one-class data, and then build one-class classifiers on learned representations. The framework not only allows to learn better representations, but also permits building one-class classifiers that are faithful to the target task. We argue that classifiers inspired by the statistical perspective in generative or discriminative models are more effective than existing approaches, such as a normality score from a surrogate classifier. We thoroughly evaluate different self-supervised representation learning algorithms under the proposed framework for one-class classification. Moreover, we present a novel distribution-augmented contrastive learning that extends training distributions via data augmentation to obstruct the uniformity of contrastive representations. In experiments, we demonstrate state-of-the-art performance on visual domain one-class classification benchmarks, including novelty and anomaly detection. Finally, we present visual explanations, confirming that the decision-making process of deep one-class classifiers is intuitive to humans. The code is available at https://github.com/google-research/deep_representation_one_class.

## 1 Introduction

One-class classification aims to identify if an example belongs to the same distribution as the training data. There are several applications of one-class classification, such as anomaly detection or outlier detection, where we learn a classifier that distinguishes the anomaly/outlier data without access to them from the normal/inlier data accessible at training. This problem is common in various domains, such as manufacturing defect detection, financial fraud detection, etc.

Generative models, such as kernel density estimation (KDE), is popular for one-class classification [1, 2] as they model the distribution by assigning high density to the training data. At test time, low density examples are determined as outliers. Unfortunately, the curse of dimensionality hinders accurate density estimation in high dimensions [3]. Deep generative models (e.g. [4, 5, 6]), have demonstrated success in modeling high-dimensional data (e.g., images) and have been applied to anomaly detection [7, 8, 9, 10, 11]. However, learning deep generative models on raw inputs remains as challenging as they appear to assign high density to background pixels [10] or learn local pixel correlations [12]. A good representation might still be beneficial to those models.

Alternately, discriminative models like one-class SVM (OC-SVM) [13] or support vector data description (SVDD) [14] learn classifiers describing the support of one-class distributions to distinguish them from outliers. These methods are powerful when being with non-linear kernels. However, its performance is still limited by the quality of input data representations.

In either generative or discriminative approaches, the fundamental limitation of one-class classification centers on learning good high-level data representations. Following the success of deep learning [15], deep one-class classifications [16, 17, 18], which extend the discriminative one-class classification using trainable deep neural networks, have shown promising results compared to their kernel counterparts. However, a naive training of deep one-class classifiers leads to a degenerate solution that maps all data into a single representation, also known as "hypersphere collapse" [16]. Previous works circumvent such issues by constraining network architectures [16], autoencoder

---

*Equal contribution.

pretraining [16, 17], surrogate multi-class classification on simulated outliers [19, 20, 21, 22] or injecting noise [18].

In this work, we present a two-stage framework for building deep one-class classifiers. As shown in Figure 1, in the first stage, we train a deep neural network to obtain a high-level data representation. In the second stage, we build a one-class classifier, such as OC-SVM or KDE, using representations from the first stage. Comparing to using surrogate losses [20, 21], our framework allows to build a classifier that is more faithful to one-class classification. Decoupling representation learning from classifier construction further opens up opportunities of using state-of-the-art representation learning methods, such as self-supervised contrastive learning [23]. While vanilla contrastive representations are less compatible with one-class classification as they are uniformly distributed on the hypersphere [24], we show that, with proper fixes, it provides representations achieving competitive one-class classification performance to previous state-of-the-arts. Furthermore, we propose a distribution-augmented contrastive learning, a novel variant of contrastive learning with distribution augmentation [25]. This is particularly effective in learning representations for one-class classification, as it reduces the class collision between examples from the same class [26] and uniformity [24]. Lastly, although representations are not optimized for one-class classification as in end-to-end trainable deep one-class classifiers [16], we demonstrate state-of-the-art performance on visual one-class classification benchmarks. We summarize our contributions as follows:

- We present a two-stage framework for building deep one-class classifiers using unsupervised and self-supervised representations followed by shallow one-class classifiers.
- We systematically study representation learning methods for one-class classification, including augmentation prediction, contrastive learning, and the proposed distribution-augmented contrastive learning method that extends training data distributions via data augmentation.
- We show that, with a good representation, both discriminative (OC-SVM) and generative (KDE) classifiers, while being competitive with each other, are better than surrogate classifiers based on the simulated outliers [20, 21].
- We achieve strong performance on visual one-class classification benchmarks, such as CIFAR-10/100 [27], Fashion MNIST [28], Cat-vs-Dog [29], CelebA [30], and MVTec AD [31].
- We extensively study the one-class contrastive learning and the realistic evaluation of anomaly detection under unsupervised and semi-supervised settings. Finally, we present visual explanations of our deep one-class classifiers to better understand their decision making processes.

## 2    RELATED WORK

One-class classification [32] has broad applications, including fraud detection [33], spam filtering [34], medical diagnosis [35], manufacturing defect detection [31], to name a few. Due to the lack of granular semantic information for one-class data, learning from unlabeled data have been employed for one-class classification. Generative models, which model the density of training data distribution, are able to determine outlier when the sample shows low density [8, 35, 36]. These include simple methods such as kernel density estimation or mixture models [37], as well as advanced ones [4, 5, 6, 38, 39, 40, 41]. However, the density from generative models for high-dimensional data could be misleading [9, 12, 42, 43]. New detection mechanisms based on the typicality [44] or likelihood ratios [10] have been proposed to improve out-of-distribution detection.

Self-supervised learning is commonly used for learning representations from unlabeled data by solving proxy tasks, such as jigsaw puzzle [45], rotation prediction [46], clustering [47], instance discrimination [48] and contrastive learning [23, 49, 50]. The learned representations are then used for multi-class classification, or transfer learning, all of which require labeled data for downstream tasks. They have also been extended to one-class classification. For example, contrastive learning is adopted to improve the out-of-distribution detection under multi-class setting [51], whereas our work focuses on learning from a single class of examples, leading to propose a novel distribution-augmented contrastive learning. Notably, learning to predict geometric transformations [20, 21, 22] extends the rotation prediction [46] to using more geometric transformations as prediction targets. Unlike typical applications of self-supervised learning where the classifier or projection head [23] are discarded after training, the geometric transformation classifier is used as a surrogate for one-class classification. As in Section 4.1, however, the surrogate classifier optimized for the self-supervised proxy task is suboptimal for one-class classification. We show that replacing it with

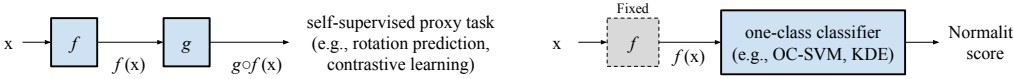

(a) Self-supervised representation learning          (b) One-class classifier

Figure 1: Overview of our two-stage framework for building deep one-class classifier. (a) In the first stage, we learn representations from one-class training distribution using self-supervised learning methods, and (b) in the second stage, we train one-class classifiers using learned representations.

| | Proxy Task | Normality Score at Test Time ($x_t$) |
|---|---|---|
| Golan and El-Yaniv [20]
Hendrycks et al. [21] | $\mathcal{L}_{\text{rot}}$ | $p_q\left(0\|f(x_t)\right)$ or $\sum_{y\in\{0,1,2,3\}} p_q\left(y\|f(\text{rot}90(x_t,y))\right)$ |
| Ours | $\mathcal{L}_{\text{rot}}, \mathcal{L}_{\text{clr}}, \mathcal{L}_{\text{clr}}^{\text{distaug}}, ...$ | $\text{KDE}\left(f(x_t)\right)$ or $\text{OC-SVM}\left(f(x_t)\right)$ |

Table 1: A comparison between one-class classifiers based on self-supervised learning. Previous works [20, 21] train one-class classifiers using augmentation prediction with geometric transformations (e.g. $\mathcal{L}_{\text{rot}}$) and determine outliers using augmentation classifiers ($p_q$). We learn representations using proxy tasks of different self-supervised learning methods, such as contrastive learning ($\mathcal{L}_{\text{clr}}$), and build simple one-class classifiers, such as KDE or OC-SVM, on learned inlier representations.

simple one-class classifiers consistently improve the performance. Furthermore, we propose strategies for better representation learning for both augmentation prediction and contrastive learning.

Distribution-augmented contrastive learning is concurrently developed in [52] as a part of their multi-task ensemble model. While sharing a similar technical formulation, we motivate from fixing the uniformity of contrastive representations. We note that our study not only focuses on representation learning, but also on the importance of detection algorithms, which is under explored before.

## 3    A TWO-STAGE FRAMEWORK FOR DEEP ONE-CLASS CLASSIFICATION

In Section 3.1, we review self-supervised representation learning algorithms, discuss how they connect to existing one-class classification methods, raise issues of state-of-the-art contrastive representation learning [23] for one-class classification, and propose ways to resolve these issues. Then, in Section 3.2, we study how to leverage the learned representations for one-class classification.

### 3.1    LEARNING REPRESENTATIONS FOR ONE-CLASS CLASSIFICATION

Let $\mathcal{A}$ be the stochastic data augmentation process, which is composed of resize and crop, horizontal flip, color jittering, gray-scale and gaussian blur, following [23], for image data. As in Figure 1, self-supervised learning methods consist of a feature extractor $f$ parameterized by deep neural networks and the proxy loss $\mathcal{L}$. Optionally, $f$ is further processed with projection head $g$ at training, which is then used to compute the proxy loss. Unless otherwise stated, $\text{normalize}(f) \triangleq f/\|f\|_2$ is used as a representation at test time. Below, we discuss details of self-supervised learning methods.

#### 3.1.1    EXTRACTING RICHER REPRESENTATION BY LEARNING WITH PROJECTION HEAD

While the efficacy of projection head has been confirmed for contrastive learning [23] or BYOL [53], it is not widely adopted for other types of self-supervised learning, including rotation prediction [46]. On the other hand, Gidaris et al. [46] show that the lower-layer representation often perform better for downstream tasks as the last layer directly involved in optimizing the proxy loss becomes overly discriminative to the proxy task, while losing useful information of the data.

Inspired by these observations, we adopt the projection head for augmentation prediction training as well. As in Figure 1a, we extend the network structure as $g \circ f$, where $g$ is the projection head used to compute proxy losses and $f$ outputs representations used for the downstream task. Note that using an identity head $g(x) = x$ recovers the network structure of previous works [20, 21, 46].

#### 3.1.2    AUGMENTATION PREDICTION

One way of representation learning is to learn by discriminating augmentations applied to the data. For example, the rotation prediction [46] learns deep representations by predicting the degree of

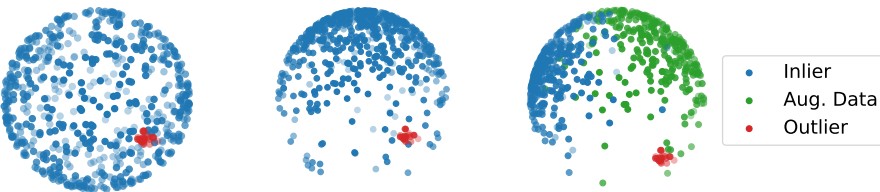

(a) Perfect uniformity.        (b) Reduced uniformity.        (c) Distribution augmentation.

Figure 2: One-class classification on different levels of uniformity of inlier distributions. (a) When representations are uniform, isolating outliers is hard. (b) Reducing uniformity makes boundary between inlier and outlier clear. (c) Distribution augmentation allows inlier distribution more compact.

rotation augmentations. The training objective of the rotation prediction task is given as follows:

$$\mathcal{L}_{\text{rot}} = \mathbb{E}_{x \sim P_{\mathcal{X}}, \mathcal{A}} \Big[ \text{CrossEntropy}\big(y, p_{q \circ g}\big(y | \text{rot90}\,(\mathcal{A}(x), y))\big)\big) \Big] \tag{1}$$

where $y \in \{0, 1, 2, 3\}$ is a prediction target representing the rotation degree, and $\text{rot90}(x, y)$ rotates an input $x$ by 90 degree $y$ times. We denote the classifier $p_{q \circ g}(y|x)$ as $p_q(y|g(x)) \propto \exp(q \circ g(x))[y]$ containing the representation $g$[1] and a linear layer $q$ with 4 output units for rotation degrees.

**Application to One-Class Classification.** Although not trained to do so, the likelihood of learned rotation classifiers[2] $p_q(y = 0|g(x))$ is shown to well approximate the normality score and has been used for one-class classification [20, 21, 22]. A plausible explanation is via outlier exposure [19], where the classifier learns a decision boundary distinguishing original images from simulated outliers by image rotation. However, it assumes inlier images are not rotated, and the classifier may not generalize to one-class classification task if it overfits to the proxy rotation prediction task.

### 3.1.3 Contrastive Learning

Unlike augmentation prediction that learns discriminative representations to data augmentation, contrastive learning [23] learns representation by distinguishing different views (e.g., augmentations) of itself from other data instances. Let $\phi(x) = \text{normalize}(g(x))$, i.e., $\|\phi(x)\| = 1$. Following [54], the proxy task loss of contrastive learning is written as:

$$\mathcal{L}_{\text{clr}} = -\mathbb{E}_{x, x_i \sim P_{\mathcal{X}}, \mathcal{A}, \mathcal{A}'} \left[ \log \frac{\exp\big(\frac{1}{\tau}\phi(\mathcal{A}(x))^\top \phi(\mathcal{A}'(x))\big)}{\exp\big(\frac{1}{\tau}\phi(\mathcal{A}(x))^\top \phi(\mathcal{A}'(x))\big) + \sum_{i=1}^{M-1} \exp\big(\frac{1}{\tau}\phi(\mathcal{A}(x))^\top \phi(\mathcal{A}(x_i))\big)} \right] \tag{2}$$

where $\mathcal{A}$ and $\mathcal{A}'$ are identical but independent stochastic augmentation processes for two different views of $x$. $\mathcal{L}_{\text{clr}}$ regularizes representations of the same instance with different views $(\mathcal{A}(x), \mathcal{A}'(x))$ to be similar, while those of different instances $(\mathcal{A}(x), \mathcal{A}'(x'))$ to be unlike.

**Class Collision and Uniformity for One-Class Classification.** While contrastive representations have achieved state-of-the-art performance on visual recognition tasks [23, 24, 49, 55] and have been theoretically proven to be effective for multi-class classification [26, 56], we argue that it could be problematic for one-class classification.

First, a class collision [26]. The contrastive loss in Eq. (2) is minimized by *maximizing* the distance between representations of negative pairs $(x, x_i), x \neq x_i$, even though they are from the same class when applied to the one-class classification. This seems to contradict to the idea of deep one-class classification [16], which learns representations by *minimizing* the distance between representations with respect to the center: $\min_{g,f} \mathbb{E}_x \|g \circ f(x) - c\|^2$.

Second, a uniformity of representations [24]. It is proved that the optimal solution for the denominator of Eq. (2) is *perfect uniformity* as $M \to \infty$ [24], meaning that $\phi(x)$ follows a uniform distribution on the hypersphere. This is problematic since one can always find an inlier $x \in \mathcal{X}$ in the proximity to any outlier $x' \notin \mathcal{X}$ on the hypersphere, as shown in Figure 2a. In contrast, with reduced uniformity as in Figure 2b, it is easier to isolate outliers from inliers.

---

[1]We abuse notation of $g$ not only to denote the projection head, but also the representation $g \circ f(\cdot)$.

[2]For presentation clarity, we use the rotation as an example for augmentations. Note that one may use more geometric transformations, such as rotation, translation, or flip of an image, as in [20, 21, 22].

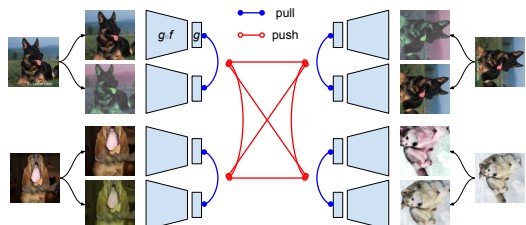

Figure 3: Distribution-augmented contrastive learning. Not only learning to discriminate different instances from an original distribution (e.g., two images of different dogs on the left), it also learns to discriminate instances from different distributions created via augmentations, such as rotations (e.g., two images of the same dog with different rotation degrees on the top).

**One-Class Contrastive Learning.** First, to reduce the uniformity of representations, we propose to use a moderate $M$ (batch size). This is in contrast with previous suggestions to train with large $M$ for contrastive representations to be most effective on multi-class classification tasks [23, 55, 57]. The impact of batch size $M$ for one-class classification will be discussed in Section 5.1.

In addition, we propose *distribution augmentation*[3] for one-class contrastive learning. The idea is that, instead of modeling the training data distribution $P_{\mathcal{X}}$, we model the union of augmented training distribution $P_{\bigcup_a a(\mathcal{X})}$, where $a(\mathcal{X}) = \{a(x)|x \in \mathcal{X}\}$. Note that augmentation $a$ for augmenting distribution is disjoint from those for data augmentation $\mathcal{A}$ that generates views. Inspired by Golan and El-Yaniv [20], we employ geometric transformations, such as rotation or horizontal flip, for distribution augmentation. For example, as in Figure 3, $x$ and $\mathrm{rot}90(x)$ (German shepherds in the top row) are considered as two separate instances and therefore are encouraged to be distant in the representation space. Not only increasing the number of data instances to train on (e.g., distribution augmentation by rotating $90°, 180°, 270°$ increases the dataset by 4 times), but it also eases the uniformity of representations on the resulted hypersphere. A pictorial example is in Figure 2c, where thanks to augmented distribution, the inlier distribution may become more compact.

## 3.2 BUILDING DEEP ONE-CLASS CLASSIFIERS WITH LEARNED REPRESENTATIONS

We present a two-stage framework for deep one-class classification that builds one-class classifiers on learned representations as in Figure 1. Compared to end-to-end training [16, 20, 21, 22], our framework provides flexibility in using various representations as the classifier is not bound to representation learning. It also allows the classifier consistent with one-class classification objective.

To construct a classifier, we revisit an old wisdom which considers the full spectrum of the distribution of the learned data representation. For generative approaches, we propose to use nonparametric kernel density estimation (KDE) to estimate densities from learned representations. For discriminative approaches, we train one-class SVMs [13]. Both methods work as a black-box and in experiment we use the default training setting except the kernel width where we reduce by 10 times than default. We provide detailed classifier formulations in Appendix A.1.

### 3.2.1 GRADIENT-BASED EXPLANATION OF DEEP ONE-CLASS CLASSIFIER

Explaining the decision making process helps users to trust deep learning models. There have been efforts to visually explain the reason for model decisions of multi-class classifiers [58, 59, 60, 61, 62, 63, 64] using the gradients computed from the classifier. In this work, we introduce a gradient-based visual explanation of one-class classification that works for any deep representations. To construct an end-to-end differentiable decision function, we employ a KDE detector, whose formulation is in Appendix A.1, built on top of any differentiable deep representations: $\frac{\partial \mathrm{KDE}(f(x))}{\partial x} = \frac{\partial \mathrm{KDE}(f(x))}{\partial f(x)} \frac{\partial f(x)}{\partial x}$.

## 4 EXPERIMENTS

Following [20], we evaluate on one-class classification benchmarks, including CIFAR-10, CIFAR-100[4] [27], Fashion-MNIST [28], and Cat-vs-Dog [29]. Images from one class are given as inlier and those from remaining classes are given as outlier. We further propose a new protocol using CelebA eyeglasses dataset [30], where face images with eyeglasses are denoted as an outlier. It is

---

[3]We note that the distribution augmentation has been applied to generative modeling [25] as a way of improving regularization and generalization via multi-task learning.

[4]20 super class labels are used for CIFAR-100 experiments [20].

| Representation | Classifier | CIFAR-10 | CIFAR-100 | f-MNIST | Cat-vs-Dog | CelebA | Mean |
|---|---|---|---|---|---|---|---|
| ResNet-50 (ImageNet) | OC-SVM | 80.0 | 83.7 | 91.8 | 74.5 | 81.4 | 84.0 |
| | KDE | 80.0 | 83.7 | 90.5 | 74.6 | 82.4 | 83.7 |
| RotNet [20] | Rotation Classifier | $86.8_{\pm0.4}$ | $80.3_{\pm0.5}$ | $87.4_{\pm1.7}$ | $86.1_{\pm0.3}$ | $51.4_{\pm3.9}$ | 83.1 |
| | KDE | $89.3_{\pm0.3}$ | $81.9_{\pm0.5}$ | $94.6_{\pm0.3}$ | $86.4_{\pm0.2}$ | $77.4_{\pm1.0}$ | 86.6 |
| Denoising | OC-SVM | $83.4_{\pm1.0}$ | $75.2_{\pm1.0}$ | $93.9_{\pm0.4}$ | $57.3_{\pm1.3}$ | $66.8_{\pm0.9}$ | 80.4 |
| | KDE | $83.5_{\pm1.0}$ | $75.2_{\pm1.0}$ | $93.7_{\pm0.4}$ | $57.3_{\pm1.3}$ | $67.0_{\pm0.7}$ | 80.4 |
| Rotation Prediction | OC-SVM | $90.8_{\pm0.3}$ | $82.8_{\pm0.6}$ | $94.6_{\pm0.3}$ | $83.7_{\pm0.6}$ | $65.8_{\pm0.9}$ | 87.1 |
| | KDE | $91.3_{\pm0.3}$ | $84.1_{\pm0.6}$ | $\mathbf{95.8}_{\pm0.3}$ | $86.4_{\pm0.6}$ | $69.5_{\pm1.7}$ | 88.2 |
| Contrastive | OC-SVM | $89.0_{\pm0.7}$ | $82.4_{\pm0.8}$ | $93.9_{\pm0.3}$ | $87.7_{\pm0.5}$ | $83.5_{\pm2.4}$ | 86.9 |
| | KDE | $89.0_{\pm0.7}$ | $82.4_{\pm0.8}$ | $93.6_{\pm0.3}$ | $87.7_{\pm0.4}$ | $84.6_{\pm2.5}$ | 86.8 |
| Contrastive (DA) | OC-SVM | $\mathbf{92.5}_{\pm0.6}$ | $\mathbf{86.5}_{\pm0.7}$ | $94.8_{\pm0.3}$ | $\mathbf{89.6}_{\pm0.5}$ | $84.5_{\pm1.1}$ | $\mathbf{89.9}$ |
| | KDE | $\mathbf{92.4}_{\pm0.7}$ | $\mathbf{86.5}_{\pm0.7}$ | $94.5_{\pm0.4}$ | $\mathbf{89.6}_{\pm0.4}$ | $\mathbf{85.6}_{\pm0.5}$ | $\mathbf{89.8}$ |

Table 2: We report the mean and standard deviation of one-class classification AUCs averaged over classes over 5 runs. The best methods are bold-faced for each setting. The per-class AUCs are reported in Appendix A.5. All methods are implemented and evaluated under the same condition.

| | CIFAR-10 | CIFAR-100 | f-MNIST | cat-vs-dog |
|---|---|---|---|---|
| Ruff et al. [16] | 64.8 | – | – | – |
| Golan and El-Yaniv [20][†] | 86.0 | 78.7 | 93.5 | 88.8 |
| Bergman and Hoshen [22][†] | 88.2 | – | 94.1 | – |
| Hendrycks et al. [21][†] | 90.1 | – | – | – |
| Huang et al. [36][†] | 86.6 | 78.8 | 93.9 | – |
| Ours: Rotation prediction | $91.3_{\pm0.3}$ | $84.1_{\pm0.6}$ | $\mathbf{95.8}_{\pm0.3}$ | $86.4_{\pm0.6}$ |
| Ours: Contrastive (DA) | $\mathbf{92.5}_{\pm0.6}$ | $\mathbf{86.5}_{\pm0.7}$ | $94.8_{\pm0.3}$ | $\mathbf{89.6}_{\pm0.5}$ |

Table 3: Comparison to previous one-class classification methods. [†] denotes evaluation methods using test time data augmentation. Our methods are both more accurate and computationally efficient.

more challenging since the difference between in and outlier samples is finer-grained. Last, in addition to semantic anomaly detection as aforementioned, we consider the defect detection benchmark MVTec [31] in Section 4.2.

We evaluate the performance of (1) representations trained with unsupervised and self-supervised learning methods, including denoising autoencoder [65], rotation prediction [46], contrastive learning [23, 48], and (2) using different one-class classifiers, such as OC-SVM or KDE. We use rotation augmentations for distribution-augmented contrastive learning, denoted as Contrastive (DA). We train a ResNet-18 [66] for feature extractor $f$ and a stack of linear, batch normalization, and ReLU, for MLP projection head $g$. More experiment details can be found in Appendix A.4.

## 4.1 MAIN RESULTS

We report the mean and standard deviation of AUCs averaged over classes over 5 runs in Table 2. The mean of 5 datasets is weighted by the number of classes for each dataset. Besides those using self-supervised representations, we provide results using ImageNet pretrained ResNet-50 to highlight the importance of learning representations from in-domain distributions.

ImageNet pretrained ResNet-50 achieves the best performance of 84.0 mean AUC over 5 datasets. Compared to representations learned with denoising objective, it works particularly well on datasets such as CIFAR-100, cat-vs-dog, and CelebA, which we attribute it to the subset of ImaegNet classes is closely related to the classes of these datasets.

Similar to the findings from [20, 21], we observe significant performance gains with self-supervised learning. Moreover, while RotNet [20], an end-to-end trained classifier using rotation prediction, achieves 83.1 AUC, the RotNet *representation* evaluated with the *KDE detector* achieves 86.6, emphasizing the importance of a proper detector in the second stage. Finally, the quality of RotNet representation improves when trained with the MLP projection head, resulting in 88.2 AUC.

The representations learned with vanilla contrastive loss achieve 86.9 AUC with OC-SVM, which under-perform those trained with rotation prediction loss (88.2). The distribution-augmented contrastive loss achieves the highest mean AUC of $\mathbf{89.9}$ among all methods, demonstrating performance gains on all datasets by a large margin upon its vanilla counterparts.

**Comparison to Previous Works.** We make comparisons to previous works in Table 3. While some comparisons may not be fair as different works use different implementations, we note that our implementation is based on the common choices of network (e.g., ResNet-18) and optimizer (e.g., momentum SGD) for image classification. We advance the previous state-of-the-art on one-class

| Protocol | RotNet [20, 21] | RotNet[†] | RotNet (MLP head)[†] | Vanilla CLR[†] | DistAug CLR[†] |
|---|---|---|---|---|---|
| Detection | $71.0_{\pm 3.5}$ | $83.5_{\pm 3.0}$ | $\mathbf{86.3}_{\pm 2.4}$ | $80.2_{\pm 1.8}$ | $\mathbf{86.5}_{\pm 1.6}$ |
| Localization | $75.6_{\pm 2.1}$ | $\mathbf{92.6}_{\pm 1.0}$ | $\mathbf{93.0}_{\pm 0.9}$ | $85.6_{\pm 1.3}$ | $90.4_{\pm 0.8}$ |

Table 4: Image-level detection and pixel-level localization AUCs on MVTec anomaly detection dataset [31]. We run experiments 5 times with different random seeds and report the mean and standard deviations. We bold-face the best entry of each row and those within the standard deviation. We use † to denote the use of KDE for an one-class classifier.

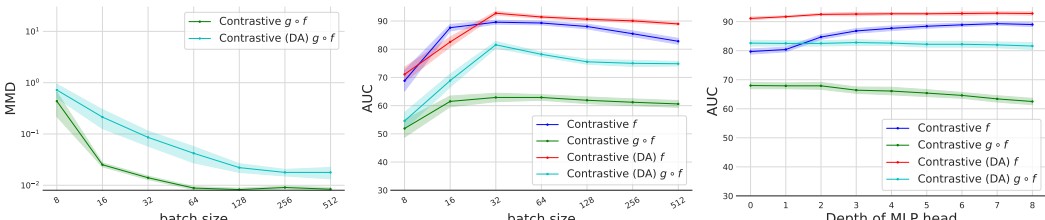

(a) MMDs with various batch sizes.  (b) AUCs with various batch sizes. (c) AUCs with various MLP depths.

Figure 4: Ablation study of contrastive representations trained with different batch sizes. We use (a) the MMD distance between the representations and the data sampled from uniform distributions to measure uniformity. Small MMD distance means being similar to uniform distribution. We also report (b) one-class classification performance in AUCs evaluated by kernel OC-SVMs and (c) the performance with MLP heads of different depths.

classification benchmarks by a large margin without test-time augmentation nor ensemble of models. We further improve the performance with model ensemble, which we report in Appendix A.2.2.

## 4.2 EXPERIMENTS ON MVTEC ANOMALY (DEFECT) DETECTION

Finally, we evaluate our proposed framework on MVTec [31] defect detection dataset, which is comprised of 15 different categories, including objects and textures. Instead of learning representations from the entire image, we learn representations of $32 \times 32$ patches. At test time, we compute the normality scores of $32 \times 32$ patches densely extracted from $256 \times 256$ images with stride of 4. For image-level detection, we apply spatial max-pooling to obtain a single image-level score. For localization, we upsample the spatial score map with Gaussian kernel [67]. Please see Appendix B for more implementation details and experimental results.

We report in Table 4 the detection and localization AUCs. We verify the similar trend to previous experiments. For example, while using the same representation, RotNet with KDE (RotNet[†]) significantly outperform the RotNet using built-in rotation classifier. Moreover, distribution-augmented contrastive learning (with rotation) improves the performance of vanilla contrastive learning. Due to a space constraint, we show localization results in Appendix B.4.

## 5 ANALYSIS AND ABLATION STUDY

In Section 5.1, we analyze behaviors of one-class contrastive representations and in Section 5.4, we report visual explanations of various deep one-class classifiers. Due to a space constraint, more studies, including an in-depth study on distribution-augmented contrastive representations and data efficiency of self-supervised learning for one-class classification, are in Appendix A.2.

### 5.1 UNIFORMITY, BATCH SIZE AND DISTRIBUTION AUGMENTATION

[23, 49, 55, 57] have shown substantial improvement on contrastive representations evaluated on multi-class classification using very large batch sizes, which results in uniformity. However, uniformity [24] and class collision [26] can be an issue for one-class classification as discussed in Section 3.1.3. Here we investigate the relations between performance and uniformity, and how we can resolve the issue via batch size, MLP head and distribution-augmented contrastive learning.

We measure the uniformity via MMD distance [68] between the learned representation and samples from uniform distributions on hyperspheres. Smaller MMD distance implies the distribution of representation is closer to uniform distributions. We train models with various batch sizes $\{2^3, \ldots, 2^9\}$. We report in Figure 4 the average and standard deviation over 5 runs on CIFAR-10 validation set.

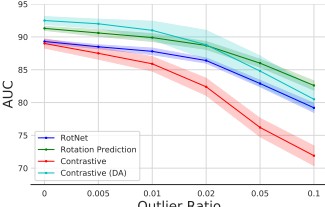 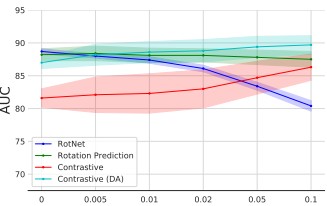 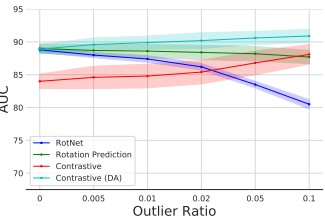

(a) Unsupervised settings for learning representation and building detector with various outlier ratio.

(b) Unsupervised setting for learning representation and 1% (50) one-class data for building detector.

(c) Unsupervised setting for learning representation and 2% (100) one-class data for building detector.

Figure 5: Realistic evaluation of anomaly detection under (5a) unsupervised and (5b, 5c) one-class or semi-supervised settings. For unsupervised settings, we either learn representation or build detector from a training set containing both inlier and outlier data without their labels. For one-class or semi-supervised settings, we learn from a training set containing only a small amount of one-class (inlier) data. We report AUCs on CIFAR-10 and OC-SVM with RBF kernel is used for evaluation.

**Distribution Augmentation.** In Figure 4a, the representations from the projection head $g \circ f$ trained by standard contrastive learning (Contrastive $g \circ f$) are closer to be uniformly distributed when we increase the batch size as proven by [24]. Therefore, in one-class classification, the nearly uniformly distributed representation from contrastive learning is only slightly better than random guess (50 AUC) as in Figure 4b. In contrast, with distribution augmentations, the learned representations (Contrastive (DA) $g \circ f$) are less uniformly distributed and result in a significant performance gain.

**Batch Size.** Following the discussion, large batch sizes result in nearly uniformly distributed representations ($g \circ f$), which is harmful to the one-class classification. On the other hand, small batch sizes ($\leq 16$), though less uniform, hinders us learning useful representations via information maximization [69]. As in Figure 4b, there is a trade-off of batch size for one-class classification, and we find batch size of 32 results in the best one-class classification performance. Further analysis showing the positive correlation between the uniformity measured by the MMD and the one-class classification performance is in Appendix A.2.2 and Figure 8.

**MLP Projection Head.** Similarly to [23], we find that $f$, an input to the projection head, performs better on the downstream task than $g \circ f$. As in Figure 4c, the performance of $f$ improves with deeper MLP head, while $g \circ f$ degrades, as it overfits to the proxy task.

Lastly, we emphasize that all these fixes contribute to an improvement of contrastive representations for one-class classification. As in Figure 4 AUC drops when any of these components are missing.

## 5.2 ANALYSIS ON DIFFERENT DISTRIBUTION AUGMENTATIONS

The choice of distributions affects the performance. The ablation study using horizontal flip (hflip) and rotation augmentations is reported in Figure 7. Note that hflip is used only to augment distribution in this experiment. Interestingly, simply adding hflip improves the AUC to 90.7. This suggests a different insight from [52] who augments distribution as a means to outlier exposure [19]. Although we report the numbers with rotation augmentations in Table 2, with hflip, rot90, rot90+hflip as augmented distributions, we achieve the best mean AUC, 93.7, on CIFAR-10, without any test-time augmentation. Additional study on distribution augmentation follows in Appendix A.2.2.

## 5.3 APPLICATIONS TO UNSUPERVISED AND SEMI-SUPERVISED ANOMALY DETECTION

We conduct experiments for unsupervised anomaly detection, where the training set may contain a few outlier data.[5] We study two settings: 1) Unsupervised settings for both learning representation and building detector, and 2) unsupervised setting for learning representation, but building detector with a small amount (as few as 50 data instances) of one-class data only. For both settings, we vary the outlier ratio in the training set from $0.5\%$ to $10\%$. We show results in Figure 5. As in Figure 5a, we observe the decrease in performance when increasing the outlier ratio as expected. Rotation prediction is slightly more robust than contrastive learning for high outlier ratio. On the other hand, when classifier is built with clean one-class data, contrastive representations performs

---

[5]We follow the definition of [70] to distinguish unsupervised and semi-supervised settings of anomaly detection. Please see Appendix A.3 for additional description on their definitions.

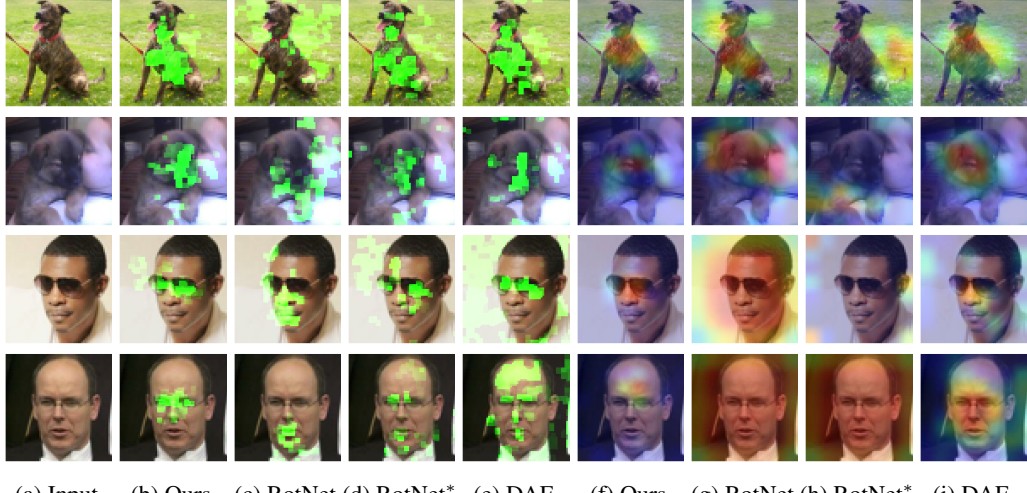

(a) Input   (b) Ours   (c) RotNet (d) RotNet* (e) DAE   (f) Ours   (g) RotNet (h) RotNet* (i) DAE

Figure 6: Visual explanations of deep one-class classifiers on cat-vs-dog and CelebA eyeglasses datasets. (a) input images, (b–e) images with heatmaps using integrated gradients [62], and (f–i) those using GradCAM [61]. RotNet*: RotNet + KDE. More examples are in Appendix A.6.

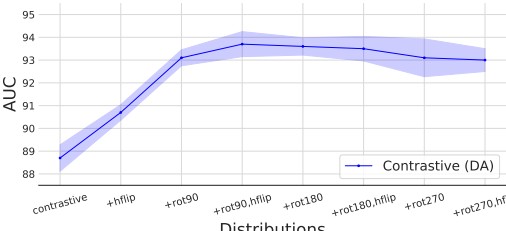

Figure 7: AUCs of contrastive representations trained from various augmented distributions on CIFAR-10. From left to right, we train by accumulating distributions. For example, an entry with "+$\mathrm{rot}90$, hflip" is trained with 4 distributions, i.e., original, hflip, $\mathrm{rot}90$ and $\mathrm{rot}90$, hflip.

better. Interestingly, contrastive learning benefits from outlier data, as it naturally learn to distinguish inlier and outlier. Due to space constraint, we provide more results and analysis in Appendix A.3.

## 5.4 VISUAL EXPLANATION OF DEEP ONE-CLASS CLASSIFIERS

We investigate the decision making process of our deep one-class classifiers using the tools described in Section 3.2.1. Specifically, we inspect by highlighting the most influential regions using two popular visual explanation algorithms, namely, integrated gradients (IG) [62] and GradCAM [61], for distribution-augmented contrastive representation, as well as RotNet and DAE on images from cat-vs-dog and CelebA eyeglasses datasets. For RotNet, we test using both a rotation classifier and KDE (RotNet* in Figure 6) to compute gradients.

As in Figure 6, the proposed visual explanation method based on the KDE one-class classifier permits highlighting human-intuitive, meaningful regions in the images, such as dog faces or eyeglasses instead of spurious background regions (Figure 6b). On the other hand, even though the classification AUC is not too worse (86.1 AUC on cat-vs-dog as opposed to 89.6 for ours), the visual explanation based on the rotation classifier (Figure 6c) suggests that the decision may be made sometimes based on the spurious features (e.g., human face in the background). We present more examples for visual explanation in Appendix A.6.

## 6 CONCLUSION

Inspired by an old wisdom of learning representations followed by building classifiers, we present a two-stage framework for deep one-class classification. We emphasize the importance of decoupling building classifiers from learning representations, which allows classifier to be consistent with the target task, one-class classification. Moreover, it permits applications of various self-supervised representation learning methods, including contrastive learning [23] with proper fixes, for one-class classification, achieving strong performance on visual one-class classification benchmarks. Finally, we exhibit visual explanation capability of our two-stage self-supervised deep one-class classifiers.

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

# A Appendix

## A.1 Formulations of One-Class Classifiers

For completeness, we provide formulations of one-class classifiers, such as one-class support vector machine (OC-SVM) [13] and kernel density estimator, used in our work.

**One-Class Support Vector Machine** solves the following optimization problem to find support vectors describing the boundary of one-class distribution:

$$\min_{w,\rho,\xi_i} \frac{\|w\|^2}{2} + \frac{1}{\nu n} \sum_{i=1}^{n} \xi_i - \rho \tag{3}$$

$$\text{subject to } w^\top f_i \geq \rho - \xi_i, \, \xi_i \geq 0, \forall i = 1, \cdots, n \tag{4}$$

where $f_i = f(x_i)$ is a feature map. The decision score is given as follows:

$$s(x) = \sum_{i=1}^{n} \alpha_i k(x_i, x) - \rho \tag{5}$$

with coefficients $\alpha_i > 0$ for support vectors. Linear or RBF ($k_\gamma(x, y) = \exp\left(-\gamma\|x - y\|^2\right)$) kernels are used for experiments.

**Kernel Density Estimation** is a nonparametric density estimation algorithm. The normality score of KDE with RBF kernel parameter $\gamma$ is written as follows:

$$\text{KDE}_\gamma(x) = \frac{1}{\gamma} \log \Big[ \sum_y \exp\left(-\gamma\|x - y\|^2\right) \Big] \tag{6}$$

## A.2 Additional Ablation Study

### A.2.1 Well Behaved Contrastive Representation

**Linear Separability.** A common belief is the good representations are linearly separable with respect to some underlying labels, which is supported by several empirical success [23]. The linear classifiers are proven to be nearly optimal [56] when we learn representations on data with *all possible labels* along with some additional assumptions. In one-class classification, although we violate the assumptions that we only train on one class of data, we are interested in the linear separability of the data to understand the characteristics of the learned embedding. To this end, in addition to training RBF kernel OC-SVM, we train OC-SVM with linear kernels on the learned embedding. We note that it is atypical to use linear OC-SVM, which is usually considered to be suboptimal for describing decision boundaries of one-class classification problems. However, with good representations, linear classifiers show good performance on one-class classification as shown in Table 5, and are better than existing works [20] in Table 2. Lastly, we note that OC-SVM with nonlinear RBF kernel is still better than linear models in Table 5. Similar observations are found in other problems such as regression [71] and generative models [72].

**Parametric Models.** In addition to studying from the perspective of discriminative model, we also dive into the generative model side. Instead of using the nonparametric KDE for density estimation, we use a parametric model, single multivariate Gaussian, whose density is defined as

$$p(x) \propto \det\left(\Sigma\right)^{-\frac{1}{2}} \exp\Big\{ - \frac{1}{2}(f(x) - \mu)^\top \Sigma^{-1}(f(x) - \mu)\Big\}. \tag{7}$$

A shown in Table 5, the Gaussian density estimation (GDE) model shows competitive performance as KDE. It suggests the learned representations from distribution-augmented contrastive learning is *compact* as our expectation, which can be well approximated by a simple parametric model with single Gaussian. Compared with nonparametric methods, although parametric models have strong assumptions on the underlying data distributions, using parametric model is more data efficient if the assumption holds[6]. In addition, parametric models have huge advantage in computation during

---

[6]The error convergence rate is $O(n^{-1/2})$, while that of nonparametric KDE is $O(n^{-1/d})$ [3].

testing. Therefore, there is trade-off between assumptions of the data as well as model flexibility and computational efficiency. We note that the single Gaussian parametric model may not be a good alternative of KDE universally. A candidate with good trade-off is Gaussian Mixture Models, which can be treated as a middle ground of nonparametric KDE and parametric single Gaussian model. We leave the study for future works.

| Representation | Classifier | CIFAR-10 | CIFAR-100 | f-MNIST | Cat-vs-Dog | CelebA | Mean |
|---|---|---|---|---|---|---|---|
| Contrastive (DA) | OC-SVM (linear) | $90.7_{\pm 0.8}$ | $81.1_{\pm 1.3}$ | $93.7_{\pm 0.8}$ | $86.3_{\pm 0.7}$ | $88.4_{\pm 1.4}$ | 86.7 |
| | OC-SVM (kernel) | $\mathbf{92.5}_{\pm 0.6}$ | $\mathbf{86.5}_{\pm 0.7}$ | $94.8_{\pm 0.3}$ | $\mathbf{89.6}_{\pm 0.5}$ | $84.5_{\pm 1.1}$ | **89.9** |
| | KDE | $92.4_{\pm 0.7}$ | $\mathbf{86.5}_{\pm 0.7}$ | $94.5_{\pm 0.4}$ | $\mathbf{89.6}_{\pm 0.4}$ | $85.6_{\pm 0.5}$ | **89.8** |
| | GDE | $92.0_{\pm 0.5}$ | $86.0_{\pm 0.8}$ | $\mathbf{95.5}_{\pm 0.3}$ | $88.9_{\pm 0.3}$ | $90.6_{\pm 0.4}$ | **89.8** |

Table 5: One-class classification results using different one-class classifiers on rotation-augmented contrastive representations. In addition to OC-SVM and KDE, both of which with RBF kernels, we test with the linear OC-SVM and the Gaussian density estimator (GDE).

### A.2.2 ANALYSIS ON DISTRIBUTION AUGMENTED CONTRASTIVE REPRESENTATIONS

**Relation to Outlier Exposure [19].** Distribution augmentation shares a similarity to outlier exposure [19] in that both methods introduce new data distributions for training. However, outlier exposure requires a stronger assumption on the data distribution that introduced outlier should not overlap with inlier distribution, while such an assumption is not required for distribution augmentation. For example, let's consider rotation prediction as an instance of outlier exposure. When training the model on randomly-rotated CIFAR-10, where we randomly rotate images of CIFAR-10, the performance of rotation prediction representation drops to 60.7 AUC, which is slightly better than a random guess. On the other hand, rotation-augmented contrastive representation is not affected by random rotation and achieves 92.4 AUC.

**Ensemble of Contrastive Representations.** We note that most previous methods have demonstrated improved performance via extensive test-time data augmentation [20, 21, 22, 52]. While already achieving state-of-the-art one-class classification performance without test-time data augmentation, we observe marginal improvement using test time data augmentation. Instead, we find that an ensemble of classifiers built on different representations significantly improve the performance. In Table 6, we report the performance of distribution-augmented contrastive representations trained with different sets of augmented distributions on CIFAR-10. We observe that ensemble of 5 models trained with different seeds consistently improves the performance. Moreover, not only we improve one-class classification AUCs when aggregating scores across models trained with different distribution augmentations, we also observe lower standard deviation across different seeds. Finally, when ensemble over $5 \times 5 = 25$ models, we achieve 94.6 AUC.

| Representations | +hflip,+rot90 | +rot90,hflip | +rot180 | +rot180,hflip | +rot270 | Ensemble of 5 |
|---|---|---|---|---|---|---|
| single model | $93.1_{\pm 0.4}$ | $93.7_{\pm 0.6}$ | $93.6_{\pm 0.4}$ | $93.5_{\pm 0.5}$ | $93.1_{\pm 0.8}$ | $\mathbf{94.4}_{\pm 0.2}$ |
| ensemble of 5 models | 93.9 | 94.4 | 94.3 | 94.2 | 93.9 | **94.6** |

Table 6: Performance of single and ensemble models of distribution augmented contrastive representations on CIFAR-10. For each augmented distribution, we report the mean and standard deviation of single model performance ("single model") and that of ensemble model whose ensemble score is aggregated from 5 models trained with different random seeds ("ensemble of 5 models"). "Ensemble of 5" aggregates score from 5 models with different augmentation distributions.

**Correlation between Uniformity and One-class Classification.** We present scatter plots in Figure 8 showing the positive correlation between the uniformity ($\log(\text{MMD})$) and the one-class classification AUCs. Specifically, we compute the MMD using $g \circ f$, an output of MLP projection head that are used to optimize the contrastive loss at training. We evaluate AUCs using both $g \circ f$ (left of Figure 8) and $f$ (right of Figure 8). Data points in the plots are obtained from vanilla and DistAug contrastive models trained with various batch sizes, including 32, 64, 128, 256 and 512. For each configuration, we train 5 models with different random seeds.

From the left plot of Figure 8, we observe a strong positive correlation between MMD and AUC, suggesting that the more uniformly distributed (i.e., lower MMD), the worse the one-class classi-

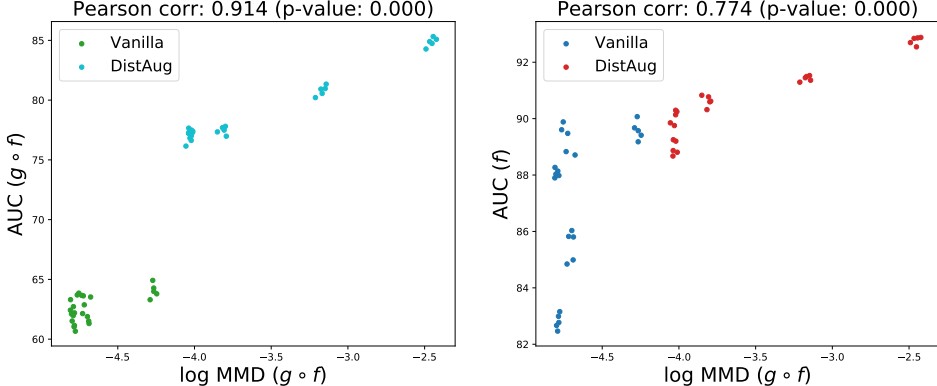

Figure 8: Scatter plots between uniformity metric ($\log(\mathrm{MMD})$) and one-class classification AUCs on CIFAR-10. MMD scores are obtained using $g \circ f$. For AUCs, we show both evaluated using $g \circ f$ (left) and $f$ (right). Data points are obtained from vanilla and DistAug contrastive representations trained with batch sizes of 32, 64, 128, 256 and 512 using 5 different random seeds.

fication performance is (i.e., low AUCs). We also compute the Pearson's correlation coefficient, which results in $0.914$. We also investigate the correlation between the MMD using $g \circ f$ and the AUC using $f$, as shown in the right plot of Figure 8. We observe highly positive correlation of $0.774$ Pearsons correlation coefficient between the MMD using $g \circ f$ and the AUC using $f$. Overall, our results suggest that reducing uniformity of $g \circ f$ improves the one-class classification accuracy tested with both $g \circ f$ and $f$ representations.

### A.2.3 DATA EFFICIENCY OF SELF-SUPERVISED REPRESENTATION LEARNING

While previous works on self-supervised learning have demonstrated the effectiveness on learning from large-scale unlabeled data, not much has shown for the data efficiency of these methods. Unlike multi-class classification tasks where the amount of data could scale multiplicative with the number of classes, data efficiency of representation learning becomes of particular interest for one-class classification as it is hard to collect large-scale data even without an annotation.

We present one-class classification AUCs of representations with various training data sizes, along with two baseline representations such as ResNet-18 using random weights or ImageNet-pretrained ResNet-50 in Figure 9. Note that we vary the training set sizes at representation learning phase only, but use a fixed amount (5000) to train classifiers for fair evaluation of the representation quality. We find that even with 50 examples, classifiers benefit from self-supervised learning when comparing against raw or deep features with random weights. The proposed distribution-augmented contrastive loss could match the performance of ImageNet-pretrained ResNet-50 with only 100 examples, while rotation prediction loss and vanilla contrastive loss require 250 and 1000 examples, respectively.

### A.3 UNSUPERVISED AND SEMI-SUPERVISED ANOMALY DETECTION

In this section, we conduct experiments for unsupervised anomaly detection. Note that one-class classification assumes access to training data drawn entirely from one-class, inlier distribution and is often referred to semi-supervised anomaly detection [70] as it requires human effort to filter out training data from outlier distribution. On the other hand, unsupervised anomaly detection assumes to include training data both from inlier and outlier distributions without knowing their respective labels. In other words, unsupervised anomaly detection may be viewed as an one-class classification with noisy data.

Here, we are interested in analyzing the impact of label noise (e.g., outlier examples are given as inlier) on one-class classification methods. To this end, we conduct experiments under unsupervised settings that includes different ratios, such as $0.5\%$, $1\%$, $2\%$, $5\%$, or $10\%$, of outlier examples in the train set without their labels. Note that the total amount of training examples for different settings remain unchanged. We show results in Figure 10 of 4 models: rotation prediction without

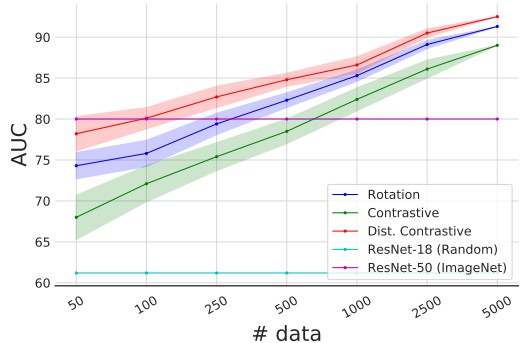

Figure 9: Self-supervised representations are trained from different data sizes, from 50 to 5000, on CIFAR-10. Standard deviations are obtained by sampling subsets 5 times. Classifiers are trained with the full (5000) train set for fair evaluation of representations. We provide baseline representations of ResNet-18 using random weights and ImageNet-pretrained ResNet-50. Standard deviations are computed by running 5 times with different seeds.

| Representation | Classifier | CIFAR-10 | CIFAR-100 | f-MNIST | cat-vs-dog | CelebA | Mean |
|---|---|---|---|---|---|---|---|
| ResNet-18 (Random) | OC-SVM (linear) | $50.1_{\pm0.9}$ | $50.1_{\pm1.7}$ | $50.3_{\pm0.9}$ | $50.0_{\pm0.4}$ | $49.9_{\pm0.3}$ | 50.1 |
| | OC-SVM (kernel) | $61.2_{\pm2.3}$ | $59.6_{\pm1.9}$ | $89.8_{\pm0.9}$ | $50.7_{\pm0.5}$ | $57.1_{\pm0.6}$ | 66.5 |
| | KDE | $60.5_{\pm2.4}$ | $58.7_{\pm2.2}$ | $89.4_{\pm1.0}$ | $50.8_{\pm0.6}$ | $57.3_{\pm0.8}$ | 65.9 |
| ResNet-50 (ImageNet) | OC-SVM (linear) | 67.9 | 71.0 | 77.0 | 61.0 | 58.0 | 70.9 |
| | OC-SVM (kernel) | 80.0 | 83.7 | 91.8 | 74.5 | 81.4 | 84.0 |
| | KDE | 80.0 | 83.7 | 90.5 | 74.6 | 82.4 | 83.7 |
| RotNet [20] | Rotation Classifier | $86.8_{\pm0.4}$ | $80.3_{\pm0.5}$ | $87.4_{\pm1.7}$ | $86.1_{\pm0.3}$ | $51.4_{\pm3.9}$ | 83.1 |
| | KDE | $89.3_{\pm0.3}$ | $81.9_{\pm0.5}$ | $94.6_{\pm0.3}$ | $86.4_{\pm0.2}$ | $77.4_{\pm1.0}$ | 86.6 |
| Denoising | OC-SVM (linear) | $72.6_{\pm1.8}$ | $62.4_{\pm2.1}$ | $55.0_{\pm3.9}$ | $61.0_{\pm1.5}$ | $59.8_{\pm4.0}$ | 62.9 |
| | OC-SVM (kernel) | $83.4_{\pm1.0}$ | $75.2_{\pm1.0}$ | $93.9_{\pm0.4}$ | $57.3_{\pm1.3}$ | $66.8_{\pm0.9}$ | 80.4 |
| | KDE | $83.5_{\pm1.0}$ | $75.2_{\pm1.0}$ | $93.7_{\pm0.4}$ | $57.3_{\pm1.3}$ | $67.0_{\pm0.7}$ | 80.4 |
| Rotation Prediction | OC-SVM (linear) | $88.5_{\pm0.8}$ | $72.3_{\pm1.7}$ | $89.0_{\pm1.5}$ | $81.4_{\pm1.4}$ | $62.6_{\pm4.1}$ | 80.1 |
| | OC-SVM (kernel) | $90.8_{\pm0.3}$ | $82.8_{\pm0.6}$ | $94.6_{\pm0.3}$ | $83.7_{\pm0.6}$ | $65.8_{\pm0.9}$ | 87.1 |
| | KDE | $91.3_{\pm0.3}$ | $84.1_{\pm0.6}$ | $\mathbf{95.8}_{\pm0.3}$ | $86.4_{\pm0.6}$ | $69.5_{\pm1.7}$ | 88.2 |
| Contrastive | OC-SVM (linear) | $85.6_{\pm1.2}$ | $74.9_{\pm1.4}$ | $92.3_{\pm1.1}$ | $82.8_{\pm1.1}$ | $\mathbf{91.7}_{\pm0.7}$ | 82.2 |
| | OC-SVM (kernel) | $89.0_{\pm0.7}$ | $82.4_{\pm0.8}$ | $93.9_{\pm0.3}$ | $87.7_{\pm0.5}$ | $83.5_{\pm2.4}$ | 86.9 |
| | KDE | $89.0_{\pm0.7}$ | $82.4_{\pm0.8}$ | $93.6_{\pm0.3}$ | $87.7_{\pm0.4}$ | $84.6_{\pm2.5}$ | 86.8 |
| Contrastive (DA) | OC-SVM (linear) | $90.7_{\pm0.8}$ | $81.1_{\pm1.3}$ | $93.7_{\pm0.8}$ | $86.3_{\pm0.7}$ | $88.4_{\pm1.4}$ | 86.7 |
| | OC-SVM (kernel) | $\mathbf{92.5}_{\pm0.6}$ | $\mathbf{86.5}_{\pm0.7}$ | $94.8_{\pm0.3}$ | $\mathbf{89.6}_{\pm0.5}$ | $84.5_{\pm1.1}$ | $\mathbf{89.9}$ |
| | KDE | $\mathbf{92.4}_{\pm0.7}$ | $\mathbf{86.5}_{\pm0.7}$ | $94.5_{\pm0.4}$ | $\mathbf{89.6}_{\pm0.4}$ | $85.6_{\pm0.5}$ | $\mathbf{89.8}$ |

Table 7: One-class classification results using different representations and one-class classifiers. We report the mean and standard deviation over 5 runs of AUCs averaged over classes. The best methods are bold-faced for each setting.

("RotNet") and with ("Rotation Prediction") MLP projection head, contrastive ("Contrastive") and distribution-augmented contrastive ("Contrastive (DA)") learning. We also report thee performance evaluated with different classifiers, including OC-SVM with RBF (Figure 10a) and linear (Figure 10c) kernels, and Gaussian Density Estimator (GDE, Figure 10b). For all models, we observe performance degradation of deep one-class classifiers trained with outlier examples as expected. Rotation prediction has shown more robust when outlier ratio is high (5 or 10%). While showing stronger performance under one-class setting, kernel-based classifiers (OC-SVM with RBF kernel, KDE) have shown less robust than parameteric (GDE) or linear classifiers under high outlier ratios, as kernel-based methods focus more on the local neighborhood structures to determine a decision boundary.

In addition, we conduct a study under another semi-supervised setting, where we are given a small amount of labeled inlier data and a large amount of unlabeled training data composed of both inlier and outlier examples. Note that this is more realistic scenario for anomaly detection problems since it is easier to obtain some portions of labeled inlier examples than outlier examples. To demonstrate

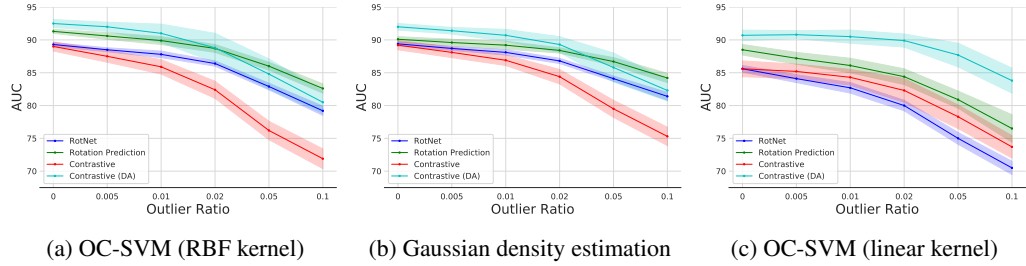

(a) OC-SVM (RBF kernel)  (b) Gaussian density estimation  (c) OC-SVM (linear kernel)

Figure 10: Classification performance under unsupervised learning setting of representation and classifier. For unsupervised setting, training set contains both inlier and outlier examples without their labels, whereas for one-class setting, training set contains only inlier examples. For representations trained with different outlier ratios, classification performances are evaluated with different classifiers, such as OC-SVMs with RBF and linear kernels, and Gaussian density estimation.

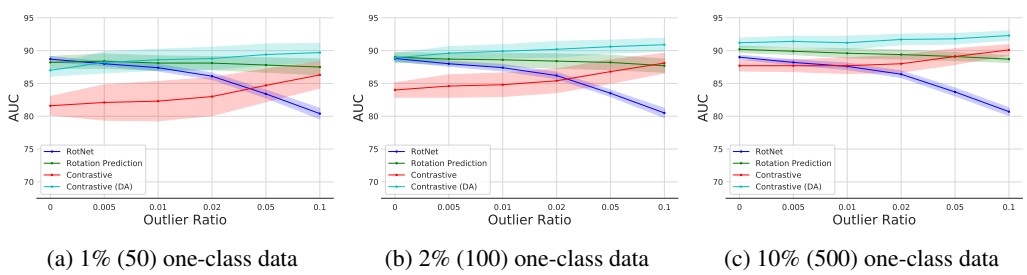

(a) 1% (50) one-class data  (b) 2% (100) one-class data  (c) 10% (500) one-class data

Figure 11: Classification performance under unsupervised representation learning and one-class classifier learning settings. For unsupervised representation learning, training set contains both inlier and outlier examples without their labels. For one-class classifier learning, a small portion of inlier data is used for training an OC-SVM with RBF kernel.

its effectiveness, we apply our proposed framework without any modification by first training representations on unlabeled training data and then building classifiers on these representations using small amount of inlier data. Results are shown in Figure 11. Interestingly, we observe consistent improvement in classification performance for contrastive representations trained on data with higher proportions of outlier examples when classifier is trained on a pure one-class data. Plausible explanation is that the model learns better contrastive representations when trained with both inlier and outlier as it naturally learn to distinguish inlier and outlier. On the other hand, representations from rotation prediction still show performance degradation as it learns to classify inlier and outlier into the same category.

## A.4 DETAILS OF EXPERIMENTAL SETTING

We resize images into $64 \times 64$ for cat-vs-dog and CelebA datasets and $32 \times 32$ for the rest. Unless otherwise stated, models are trained for 2048 epochs with momentum (0.9) SGD and a single cycle of cosine learning rate decay [73]. L2 weight regularization with coefficient of 0.0003 is applied. We use scikit-learn [74] implementation of OC-SVMs with default value of $\nu$. We use $\gamma = {}^{10}/_{|f| \cdot \mathrm{Var}(f)}$, which is 10 times larger than the default value for kernel OC-SVM. Same value of $\gamma$ is used for KDE. We use scikit-learn implementation of Gaussian mixture model using a single component for Gaussian density estimator (GDE). No hyperparameters are tuned for GDE. Finally, all experiments are conducted using TensorFlow [75].

Other hyperparameters, such as learning rate or the MLP projection head depth, are cross-validated using small labeled data. While this may violate the assumption of one-class classification, supervised model selection is inevitable for deep learning models as their behaviors may be largely dependent on hyperparameters. To this end, we use 10% of inlier (500) and the same number of outlier examples of CIFAR-10 for hyperparameter selection, and use the same set of hyperparameters to test methods on other datasets, which could demonstrate the algorithm robustness. Learning rate

$\in\{0.1, 0.03, 0.01, 0.003, 0.001\}$ and the depth $\in\{0, \cdots, 8\}$ of an MLP projection head are tuned for all methods. In addition, the temperature $\tau\in\{1, 0.5, 0.2, 0.1\}$ and the batch size $\in\{2^n, n = 3, ..., 9\}$ of contrastive loss are tuned. To this end, we train all models across all datasets using the same hyper-parameter configurations, such as learning rate of $0.01$, projection head of depth $8$ ($[512\times8, 128]$), temperature $\tau$ of $0.2$, or batch size of $32$.

## A.5 PER-CLASS AUCS

| Representation | RotNet [20] | | Denoising | | Rot. Prediction | | Contrastive | | Contrastive (DA) | |
|---|---|---|---|---|---|---|---|---|---|---|
| Classifier | Rot. Cls | KDE | OC-SVM | KDE | OC-SVM | KDE | OC-SVM | KDE | OC-SVM | KDE |
| airplane | $80.3_{\pm0.4}$ | $81.4_{\pm0.3}$ | $81.6_{\pm0.7}$ | $81.6_{\pm0.7}$ | $83.6_{\pm0.6}$ | $88.5_{\pm0.3}$ | $88.8_{\pm0.3}$ | $88.8_{\pm0.2}$ | $90.9_{\pm0.5}$ | $91.0_{\pm0.5}$ |
| automobile | $91.2_{\pm0.3}$ | $96.7_{\pm0.1}$ | $92.4_{\pm0.4}$ | $92.4_{\pm0.4}$ | $96.9_{\pm0.2}$ | $97.5_{\pm0.3}$ | $97.5_{\pm0.2}$ | $97.5_{\pm0.2}$ | $98.9_{\pm0.1}$ | $98.9_{\pm0.1}$ |
| bird | $85.3_{\pm0.3}$ | $86.4_{\pm0.3}$ | $75.9_{\pm1.3}$ | $75.9_{\pm1.3}$ | $87.9_{\pm0.4}$ | $88.2_{\pm0.3}$ | $87.7_{\pm0.4}$ | $87.7_{\pm0.4}$ | $88.1_{\pm0.1}$ | $88.0_{\pm0.2}$ |
| cat | $78.1_{\pm0.3}$ | $78.9_{\pm0.3}$ | $72.3_{\pm1.9}$ | $72.3_{\pm1.9}$ | $79.0_{\pm0.6}$ | $78.3_{\pm1.0}$ | $82.0_{\pm0.4}$ | $82.1_{\pm0.4}$ | $83.1_{\pm0.8}$ | $83.2_{\pm0.9}$ |
| deer | $85.9_{\pm0.5}$ | $88.0_{\pm0.6}$ | $82.3_{\pm0.6}$ | $82.4_{\pm0.6}$ | $90.5_{\pm0.2}$ | $90.2_{\pm0.2}$ | $82.4_{\pm1.8}$ | $82.3_{\pm1.8}$ | $89.9_{\pm1.3}$ | $89.4_{\pm1.8}$ |
| dog | $86.7_{\pm0.4}$ | $88.1_{\pm0.3}$ | $83.1_{\pm0.6}$ | $83.1_{\pm0.6}$ | $89.5_{\pm0.4}$ | $88.2_{\pm0.6}$ | $89.2_{\pm0.4}$ | $89.2_{\pm0.5}$ | $90.3_{\pm1.0}$ | $90.2_{\pm1.2}$ |
| frog | $89.6_{\pm0.5}$ | $90.7_{\pm0.4}$ | $86.7_{\pm0.6}$ | $86.8_{\pm0.6}$ | $94.1_{\pm0.3}$ | $94.6_{\pm0.3}$ | $89.7_{\pm1.4}$ | $89.8_{\pm1.4}$ | $93.5_{\pm0.6}$ | $93.5_{\pm0.6}$ |
| horse | $93.3_{\pm0.6}$ | $95.7_{\pm0.3}$ | $91.2_{\pm0.4}$ | $91.2_{\pm0.4}$ | $96.7_{\pm0.1}$ | $97.0_{\pm0.1}$ | $95.6_{\pm0.2}$ | $95.6_{\pm0.2}$ | $98.2_{\pm0.1}$ | $98.1_{\pm0.1}$ |
| ship | $91.8_{\pm0.5}$ | $94.0_{\pm0.2}$ | $78.0_{\pm2.6}$ | $78.0_{\pm2.6}$ | $95.0_{\pm0.2}$ | $95.7_{\pm0.1}$ | $86.0_{\pm0.7}$ | $85.8_{\pm0.6}$ | $96.5_{\pm0.3}$ | $96.5_{\pm0.3}$ |
| truck | $86.0_{\pm0.6}$ | $93.0_{\pm0.3}$ | $91.0_{\pm0.6}$ | $91.0_{\pm0.6}$ | $94.9_{\pm0.2}$ | $94.9_{\pm0.2}$ | $90.6_{\pm0.9}$ | $90.7_{\pm0.9}$ | $95.2_{\pm1.3}$ | $95.1_{\pm1.3}$ |
| mean | $86.8_{\pm0.4}$ | $89.3_{\pm0.3}$ | $83.4_{\pm1.0}$ | $83.5_{\pm1.0}$ | $90.8_{\pm0.3}$ | $91.3_{\pm0.3}$ | $89.0_{\pm0.7}$ | $89.0_{\pm0.7}$ | $92.5_{\pm0.6}$ | $92.4_{\pm0.7}$ |

Table 8: Per-class one-class classification AUCs on CIFAR-10.

| Representation | RotNet [20] | | Denoising | | Rot. Prediction | | Contrastive | | Contrastive (DA) | |
|---|---|---|---|---|---|---|---|---|---|---|
| Classifier | Rot. Cls | KDE | OC-SVM | KDE | OC-SVM | KDE | OC-SVM | KDE | OC-SVM | KDE |
| 0 | $77.7_{\pm0.6}$ | $79.1_{\pm0.6}$ | $74.6_{\pm0.7}$ | $74.4_{\pm0.8}$ | $77.1_{\pm0.8}$ | $78.8_{\pm0.6}$ | $79.9_{\pm1.0}$ | $79.9_{\pm1.0}$ | $83.0_{\pm1.4}$ | $82.9_{\pm1.4}$ |
| 1 | $73.0_{\pm0.9}$ | $75.4_{\pm0.8}$ | $71.8_{\pm1.6}$ | $71.8_{\pm1.7}$ | $75.2_{\pm0.9}$ | $77.9_{\pm1.1}$ | $81.1_{\pm0.7}$ | $81.1_{\pm0.8}$ | $84.6_{\pm0.3}$ | $84.3_{\pm0.4}$ |
| 2 | $72.9_{\pm0.6}$ | $76.1_{\pm0.7}$ | $81.4_{\pm0.5}$ | $81.4_{\pm0.5}$ | $84.4_{\pm0.3}$ | $85.7_{\pm0.4}$ | $87.4_{\pm0.4}$ | $87.5_{\pm0.4}$ | $88.5_{\pm0.6}$ | $88.6_{\pm0.6}$ |
| 3 | $79.6_{\pm0.6}$ | $81.3_{\pm0.3}$ | $77.6_{\pm1.6}$ | $77.6_{\pm1.6}$ | $85.1_{\pm0.6}$ | $87.3_{\pm0.4}$ | $84.7_{\pm0.8}$ | $84.7_{\pm0.8}$ | $86.3_{\pm0.8}$ | $86.4_{\pm0.8}$ |
| 4 | $78.4_{\pm0.4}$ | $80.8_{\pm0.5}$ | $76.1_{\pm1.5}$ | $76.1_{\pm1.5}$ | $81.1_{\pm0.8}$ | $84.8_{\pm0.8}$ | $89.8_{\pm0.6}$ | $89.9_{\pm0.6}$ | $92.6_{\pm0.3}$ | $92.6_{\pm0.3}$ |
| 5 | $73.1_{\pm0.9}$ | $73.7_{\pm0.5}$ | $70.2_{\pm0.5}$ | $70.2_{\pm0.5}$ | $72.0_{\pm1.3}$ | $78.6_{\pm1.0}$ | $82.6_{\pm1.2}$ | $82.7_{\pm1.2}$ | $84.4_{\pm1.3}$ | $84.5_{\pm1.3}$ |
| 6 | $89.8_{\pm0.5}$ | $90.4_{\pm0.4}$ | $68.5_{\pm2.4}$ | $68.4_{\pm2.3}$ | $88.6_{\pm0.3}$ | $87.5_{\pm0.4}$ | $77.0_{\pm1.6}$ | $77.0_{\pm1.6}$ | $73.4_{\pm0.9}$ | $73.4_{\pm0.9}$ |
| 7 | $64.1_{\pm0.7}$ | $66.1_{\pm0.6}$ | $77.6_{\pm0.4}$ | $77.6_{\pm0.4}$ | $73.0_{\pm1.0}$ | $76.2_{\pm0.6}$ | $80.5_{\pm0.3}$ | $80.4_{\pm0.3}$ | $84.3_{\pm0.6}$ | $84.2_{\pm0.6}$ |
| 8 | $83.5_{\pm0.6}$ | $85.3_{\pm0.4}$ | $81.7_{\pm0.6}$ | $81.7_{\pm0.6}$ | $87.8_{\pm0.1}$ | $86.4_{\pm0.2}$ | $85.2_{\pm0.6}$ | $85.2_{\pm0.6}$ | $87.5_{\pm0.9}$ | $87.7_{\pm0.9}$ |
| 9 | $91.4_{\pm0.2}$ | $91.7_{\pm0.4}$ | $73.7_{\pm1.1}$ | $73.7_{\pm1.1}$ | $88.0_{\pm0.7}$ | $88.8_{\pm0.5}$ | $89.1_{\pm0.9}$ | $89.2_{\pm0.9}$ | $94.1_{\pm0.3}$ | $94.1_{\pm0.3}$ |
| 10 | $86.1_{\pm0.6}$ | $87.0_{\pm0.6}$ | $58.6_{\pm2.2}$ | $58.5_{\pm2.2}$ | $79.5_{\pm1.1}$ | $82.4_{\pm0.7}$ | $60.0_{\pm2.0}$ | $59.8_{\pm2.0}$ | $85.1_{\pm1.1}$ | $85.2_{\pm1.1}$ |
| 11 | $83.7_{\pm0.5}$ | $85.0_{\pm0.3}$ | $76.5_{\pm0.7}$ | $76.5_{\pm0.7}$ | $87.2_{\pm0.2}$ | $84.9_{\pm0.4}$ | $84.3_{\pm0.2}$ | $84.2_{\pm0.1}$ | $87.8_{\pm0.6}$ | $87.8_{\pm0.6}$ |
| 12 | $82.8_{\pm0.4}$ | $84.3_{\pm0.4}$ | $79.0_{\pm1.1}$ | $79.1_{\pm1.1}$ | $85.8_{\pm0.3}$ | $85.2_{\pm0.7}$ | $84.4_{\pm0.5}$ | $84.4_{\pm0.5}$ | $82.0_{\pm1.0}$ | $82.0_{\pm1.1}$ |
| 13 | $61.8_{\pm1.2}$ | $64.1_{\pm0.8}$ | $70.2_{\pm0.7}$ | $70.2_{\pm0.7}$ | $70.8_{\pm0.7}$ | $73.6_{\pm0.6}$ | $79.3_{\pm1.4}$ | $79.2_{\pm1.4}$ | $82.7_{\pm0.4}$ | $82.7_{\pm0.4}$ |
| 14 | $89.7_{\pm0.3}$ | $90.3_{\pm0.2}$ | $73.1_{\pm1.0}$ | $73.1_{\pm1.0}$ | $91.3_{\pm0.3}$ | $90.2_{\pm0.6}$ | $89.1_{\pm0.7}$ | $89.1_{\pm0.6}$ | $93.4_{\pm0.2}$ | $93.4_{\pm0.2}$ |
| 15 | $69.4_{\pm0.4}$ | $70.6_{\pm0.3}$ | $71.2_{\pm0.7}$ | $71.1_{\pm0.7}$ | $73.4_{\pm0.5}$ | $74.9_{\pm0.5}$ | $71.5_{\pm1.2}$ | $71.4_{\pm1.2}$ | $76.1_{\pm1.3}$ | $75.8_{\pm1.4}$ |
| 16 | $78.0_{\pm0.3}$ | $79.4_{\pm0.4}$ | $74.7_{\pm1.0}$ | $74.8_{\pm1.0}$ | $79.8_{\pm0.9}$ | $80.5_{\pm0.9}$ | $79.3_{\pm0.2}$ | $79.3_{\pm0.2}$ | $80.4_{\pm0.7}$ | $80.3_{\pm0.7}$ |
| 17 | $92.5_{\pm0.4}$ | $93.7_{\pm0.2}$ | $88.8_{\pm0.5}$ | $88.8_{\pm0.5}$ | $93.9_{\pm0.3}$ | $94.6_{\pm0.2}$ | $91.2_{\pm0.3}$ | $91.2_{\pm0.3}$ | $97.5_{\pm0.1}$ | $97.5_{\pm0.1}$ |
| 18 | $90.8_{\pm0.3}$ | $92.5_{\pm0.3}$ | $80.3_{\pm0.7}$ | $80.2_{\pm0.7}$ | $93.4_{\pm0.3}$ | $93.1_{\pm0.1}$ | $87.4_{\pm0.3}$ | $87.2_{\pm0.3}$ | $94.4_{\pm0.2}$ | $94.4_{\pm0.2}$ |
| 19 | $88.4_{\pm0.4}$ | $90.3_{\pm0.3}$ | $78.0_{\pm0.8}$ | $77.8_{\pm0.8}$ | $89.3_{\pm0.1}$ | $89.8_{\pm0.2}$ | $84.0_{\pm1.6}$ | $83.9_{\pm1.6}$ | $92.5_{\pm0.7}$ | $92.4_{\pm0.6}$ |
| mean | $80.3_{\pm0.5}$ | $81.9_{\pm0.5}$ | $75.2_{\pm1.0}$ | $75.2_{\pm1.0}$ | $82.8_{\pm0.6}$ | $84.1_{\pm0.6}$ | $82.4_{\pm0.8}$ | $82.4_{\pm0.8}$ | $86.5_{\pm0.7}$ | $86.5_{\pm0.7}$ |

Table 9: Per-class one-class classification AUCs on CIFAR-20.

| Representation | RotNet [20] | | Denoising | | Rot. Prediction | | Contrastive | | Contrastive (DA) | |
|---|---|---|---|---|---|---|---|---|---|---|
| Classifier | Rot. Cls | KDE | OC-SVM | KDE | OC-SVM | KDE | OC-SVM | KDE | OC-SVM | KDE |
| 0 | $83.7_{\pm2.2}$ | $94.5_{\pm0.3}$ | $91.8_{\pm0.7}$ | $92.3_{\pm0.7}$ | $94.5_{\pm0.3}$ | $95.2_{\pm0.5}$ | $93.4_{\pm0.2}$ | $93.3_{\pm0.2}$ | $93.0_{\pm0.9}$ | $92.3_{\pm1.6}$ |
| 1 | $96.2_{\pm1.9}$ | $99.5_{\pm0.1}$ | $97.4_{\pm0.4}$ | $97.7_{\pm0.4}$ | $99.4_{\pm0.1}$ | $99.7_{\pm0.1}$ | $98.1_{\pm0.2}$ | $98.3_{\pm0.2}$ | $99.2_{\pm0.4}$ | $99.1_{\pm0.6}$ |
| 2 | $78.1_{\pm2.0}$ | $93.8_{\pm0.2}$ | $91.6_{\pm0.4}$ | $91.6_{\pm0.4}$ | $93.1_{\pm0.3}$ | $94.2_{\pm0.3}$ | $93.3_{\pm0.2}$ | $93.5_{\pm0.2}$ | $93.1_{\pm0.2}$ | $93.4_{\pm0.1}$ |
| 3 | $83.7_{\pm2.1}$ | $95.0_{\pm0.4}$ | $91.1_{\pm1.1}$ | $91.0_{\pm1.1}$ | $93.3_{\pm0.5}$ | $95.4_{\pm0.4}$ | $92.2_{\pm0.5}$ | $91.4_{\pm0.7}$ | $93.2_{\pm0.3}$ | $92.2_{\pm0.3}$ |
| 4 | $83.5_{\pm2.1}$ | $91.9_{\pm0.1}$ | $93.3_{\pm0.2}$ | $92.7_{\pm0.2}$ | $92.1_{\pm0.2}$ | $93.9_{\pm0.2}$ | $93.4_{\pm0.1}$ | $92.5_{\pm0.2}$ | $94.2_{\pm0.2}$ | $93.4_{\pm0.3}$ |
| 5 | $87.6_{\pm1.5}$ | $94.3_{\pm0.4}$ | $94.9_{\pm0.4}$ | $94.7_{\pm0.5}$ | $95.1_{\pm0.3}$ | $97.3_{\pm0.3}$ | $93.8_{\pm0.1}$ | $93.0_{\pm0.2}$ | $94.3_{\pm0.2}$ | $94.3_{\pm0.3}$ |
| 6 | $77.3_{\pm1.2}$ | $83.4_{\pm0.2}$ | $86.1_{\pm0.5}$ | $85.5_{\pm0.4}$ | $82.5_{\pm0.4}$ | $85.9_{\pm0.4}$ | $88.3_{\pm0.1}$ | $87.4_{\pm0.1}$ | $87.6_{\pm0.2}$ | $86.2_{\pm0.1}$ |
| 7 | $96.9_{\pm0.8}$ | $99.2_{\pm0.1}$ | $96.9_{\pm0.3}$ | $96.9_{\pm0.3}$ | $98.7_{\pm0.1}$ | $99.4_{\pm0.1}$ | $97.3_{\pm0.2}$ | $97.0_{\pm0.2}$ | $98.5_{\pm0.1}$ | $98.4_{\pm0.1}$ |
| 8 | $92.2_{\pm1.5}$ | $96.2_{\pm0.5}$ | $90.9_{\pm0.8}$ | $90.0_{\pm0.9}$ | $98.5_{\pm0.3}$ | $98.3_{\pm0.3}$ | $92.3_{\pm1.2}$ | $92.3_{\pm1.2}$ | $97.6_{\pm0.3}$ | $97.6_{\pm0.3}$ |
| 9 | $95.0_{\pm1.3}$ | $98.1_{\pm0.3}$ | $93.3_{\pm0.5}$ | $92.9_{\pm0.5}$ | $99.0_{\pm0.1}$ | $98.6_{\pm0.2}$ | $96.9_{\pm0.3}$ | $96.9_{\pm0.3}$ | $97.9_{\pm0.3}$ | $98.0_{\pm0.3}$ |
| mean | $87.4_{\pm1.7}$ | $94.6_{\pm0.3}$ | $92.7_{\pm0.5}$ | $92.5_{\pm0.5}$ | $94.6_{\pm0.3}$ | $95.8_{\pm0.3}$ | $93.9_{\pm0.3}$ | $93.6_{\pm0.3}$ | $94.8_{\pm0.3}$ | $94.5_{\pm0.4}$ |

Table 10: Per-class one-class classification AUCs on Fashion MNIST.

| Representation | RotNet [20] | | Denoising | | Rot. Prediction | | Contrastive | | Contrastive (DA) | |
|---|---|---|---|---|---|---|---|---|---|---|
| Classifier | Rot. Cls | KDE | OC-SVM | KDE | OC-SVM | KDE | OC-SVM | KDE | OC-SVM | KDE |
| cat | $86.4_{\pm0.3}$ | $86.1_{\pm0.2}$ | $41.3_{\pm1.7}$ | $41.2_{\pm1.8}$ | $81.9_{\pm0.7}$ | $84.1_{\pm0.8}$ | $89.8_{\pm0.4}$ | $89.7_{\pm0.4}$ | $91.6_{\pm0.3}$ | $91.7_{\pm0.3}$ |
| dog | $85.8_{\pm0.3}$ | $86.6_{\pm0.2}$ | $60.6_{\pm1.1}$ | $60.3_{\pm1.0}$ | $85.5_{\pm0.6}$ | $88.7_{\pm0.4}$ | $85.7_{\pm0.5}$ | $85.7_{\pm0.5}$ | $87.5_{\pm0.6}$ | $87.5_{\pm0.6}$ |
| mean | $86.1_{\pm0.3}$ | $86.4_{\pm0.2}$ | $51.0_{\pm1.4}$ | $50.7_{\pm1.4}$ | $83.7_{\pm0.6}$ | $86.4_{\pm0.6}$ | $87.7_{\pm0.5}$ | $87.7_{\pm0.4}$ | $89.6_{\pm0.5}$ | $89.6_{\pm0.4}$ |

Table 11: Per-class one-class classification AUCs on Cat-vs-Dog.

## A.6 MORE EXAMPLES FOR VISUAL EXPLANATION OF DEEP ONE-CLASS CLASSIFIERS

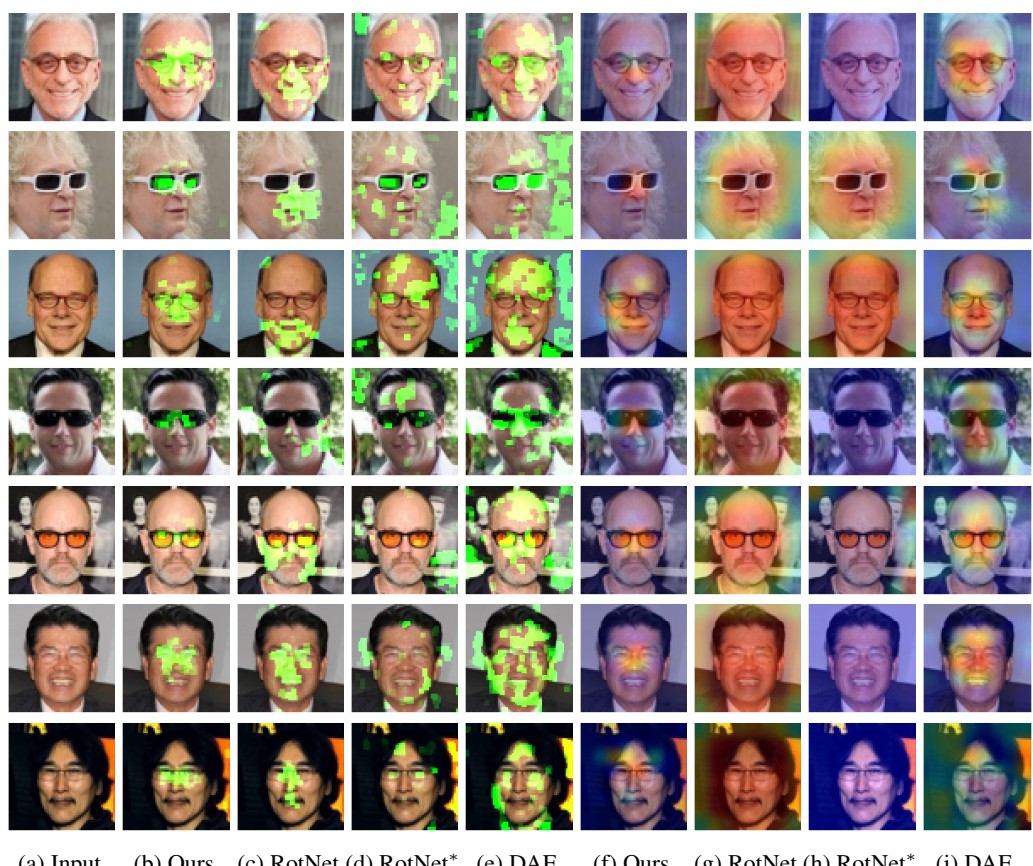

(a) Input    (b) Ours    (c) RotNet   (d) RotNet*   (e) DAE    (f) Ours    (g) RotNet   (h) RotNet*   (i) DAE

Figure 12: Visual explanations on CelebA eyeglasses dataset. (a) input images, (b–e) images with heatmaps using integrated gradients [62], and (f–i) those using GradCAM [61]. RotNet*: RotNet + KDE.

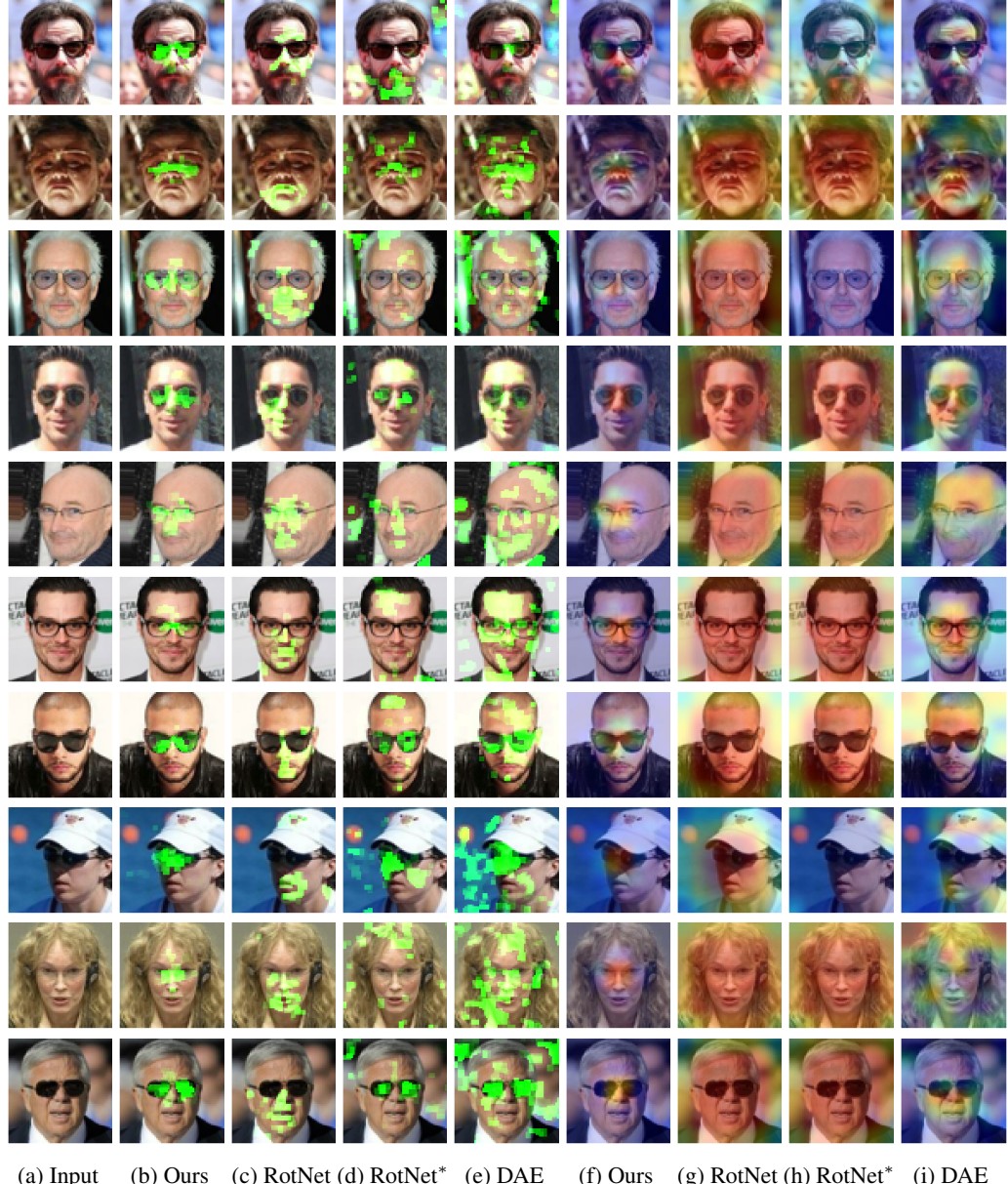

(a) Input    (b) Ours    (c) RotNet (d) RotNet* (e) DAE    (f) Ours    (g) RotNet (h) RotNet* (i) DAE

Figure 13: Visual explanations on CelebA eyeglasses dataset. (a) input images, (b–e) images with heatmaps using integrated gradients [62], and (f–i) those using GradCAM [61]. RotNet*: RotNet + KDE.

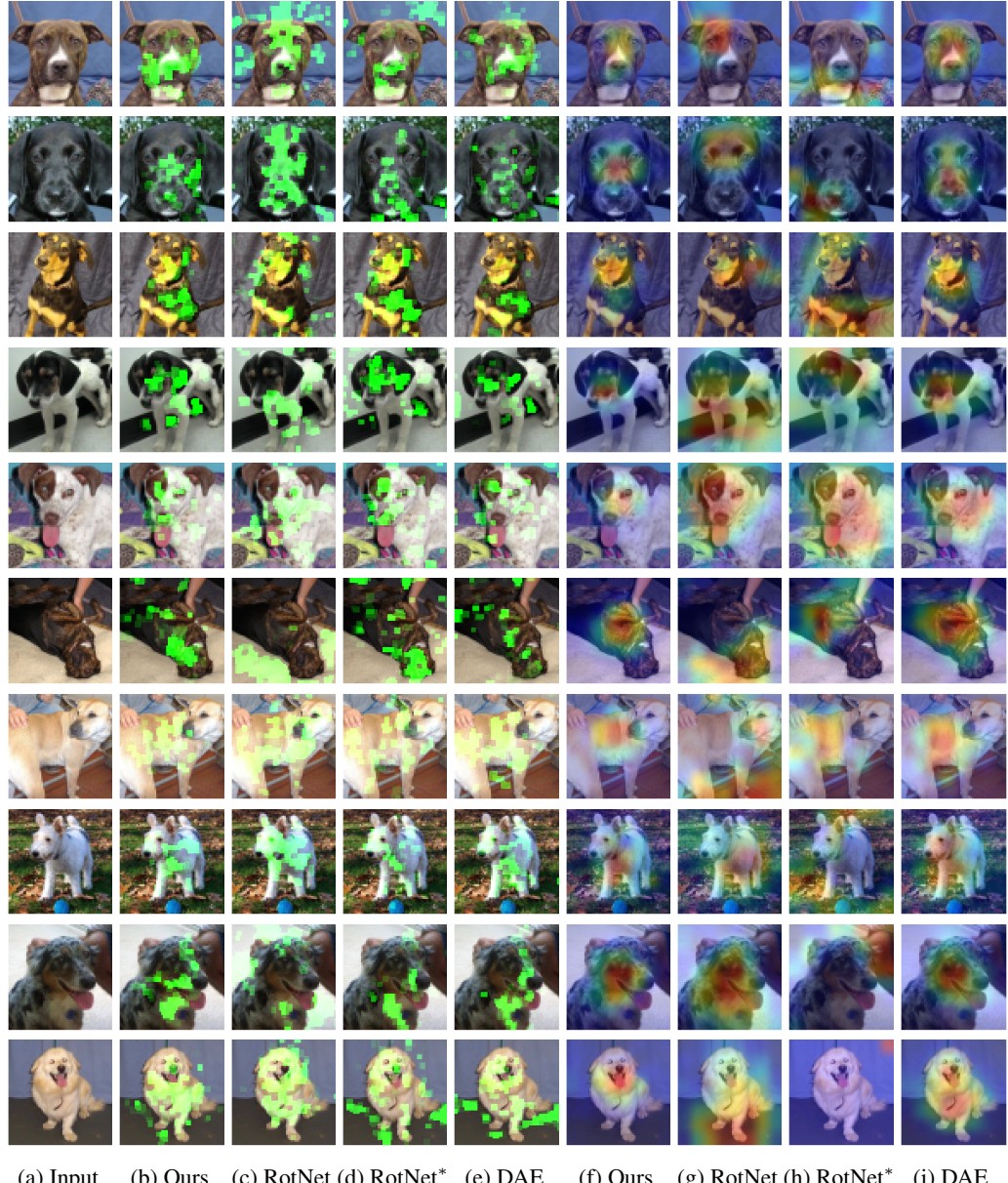

(a) Input    (b) Ours    (c) RotNet    (d) RotNet*    (e) DAE    (f) Ours    (g) RotNet    (h) RotNet*    (i) DAE

Figure 14: Visual explanations on cat-vs-dog dataset. (a) input images, (b–e) images with heatmaps using integrated gradients [62], and (f–i) those using GradCAM [61]. RotNet*: RotNet + KDE.

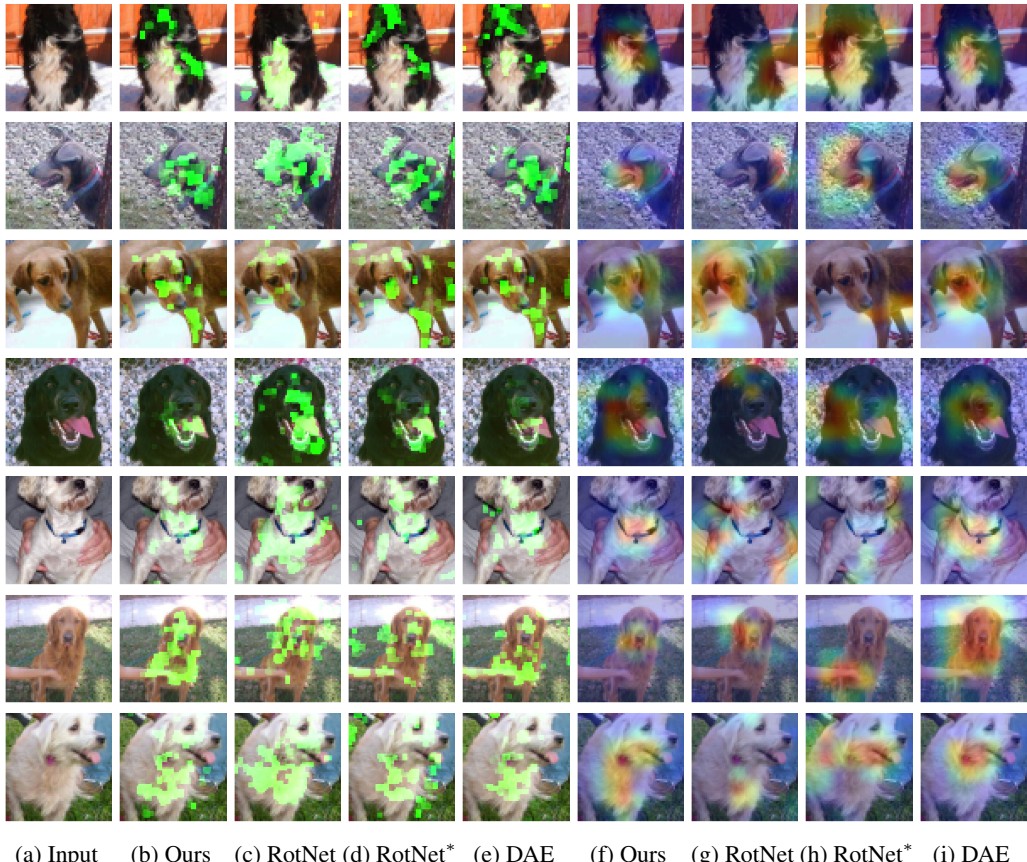

(a) Input     (b) Ours     (c) RotNet (d) RotNet* (e) DAE     (f) Ours     (g) RotNet (h) RotNet* (i) DAE

Figure 15: Visual explanations on cat-vs-dog dataset. (a) input images, (b–e) images with heatmaps using integrated gradients [62], and (f–i) those using GradCAM [61]. RotNet*: RotNet + KDE.

# B  Experiments on MVTec Anomaly Detection Dataset

## B.1  Dataset Description

There are 15 different categories in MVTec anomaly detection dataset [31], where 10 are object (e.g., bottle, cable, transistor) and 5 are texture (e.g., grid, leather) categories. Each category comes with the training set containing $\sim$241 images per category from normal data distribution and the test set containing $\sim$115 images per category from both normal and defective data distributions. The dataset also provides pixel-accurate annotation for defective regions, allowing to measure the defect localization performance. Images are of high-resolution, whose side length is as long as 1024.

## B.2  Experimental Setting

First, we resize all images into 256$\times$256 in our experiments. One challenge of learning representations from scratch on MVTec dataset is that the amount of training set is too small ($\sim$241 images per category). As in Figure 9, self-supervised representation learning methods benefit from the large amount of training data and the performance degrades when the amount of training data is limited. Instead of learning the holistic image representation, we learn a patch representation of size 32$\times$32 via the proposed self-supervised representation learning methods. Similarly to our previous experiments, we train ResNet-18 from scratch.

For evaluation, we compute both image-level detection and localization AUCs. We densely extract embeddings $f$ from image patches with the stride of 4 and compute anomaly scores with KDE at each patch location, which results in a $(\frac{256-224}{4}+1)\times(\frac{256-224}{4}+1)=57\times57$ anomaly score map. For image-level detection, we apply spatial max-pooling to obtain a single score. For localization, we upsample 57$\times$57 into 256$\times$256 with Gaussian kernel[7] [67] and compare with the annotation.

## B.3  Experimental Results

| AUC | CAVGA-$R_u$ [76] | RotNet [20, 21] | RotNet + KDE | RotNet (MLP head) + KDE | Vanilla Contrastive | DistAug Contrastive |
|---|---|---|---|---|---|---|
| Detection on object | 83.8 | $77.9_{\pm2.3}$ | $85.9_{\pm2.1}$ | $\mathbf{89.0}_{\pm2.0}$ | $83.8_{\pm1.4}$ | $\mathbf{88.6}_{\pm1.4}$ |
| Detection on texture | 78.2 | $73.2_{\pm3.5}$ | $75.5_{\pm3.1}$ | $\mathbf{81.0}_{\pm3.4}$ | $73.0_{\pm2.6}$ | $\mathbf{82.5}_{\pm2.2}$ |
| Detection on all | 81.9 | $71.0_{\pm3.5}$ | $83.5_{\pm3.0}$ | $\mathbf{86.3}_{\pm2.4}$ | $80.2_{\pm1.8}$ | $\mathbf{86.5}_{\pm1.6}$ |
| Localization on object | – | $74.1_{\pm1.6}$ | $95.7_{\pm0.7}$ | $\mathbf{96.4}_{\pm0.4}$ | $91.7_{\pm1.0}$ | $94.4_{\pm0.5}$ |
| Localization on texture | – | $78.8_{\pm3.1}$ | $\mathbf{86.4}_{\pm1.8}$ | $86.3_{\pm2.0}$ | $73.4_{\pm1.8}$ | $82.5_{\pm1.5}$ |
| Localization on all | 89 | $75.6_{\pm2.1}$ | $\mathbf{92.6}_{\pm1.0}$ | $\mathbf{93.0}_{\pm0.9}$ | $85.6_{\pm1.3}$ | $90.4_{\pm0.8}$ |

Table 12: Image-level detection and pixel-level localization AUC results on MVTec anomaly detection dataset. We run experiments 5 times with different random seeds and report the mean and standard deviations. We bold-face the best entry of each row and those within the standard deviation.

We report quantitative results in Table 12. Specifically, we report detection and localization AUCs averaged over object, texture, and all categories. Likewise, we run experiments 5 times with different random seeds and report the mean and standard deviation. We report the performance of RotNet (using rotation classifier, as in [20, 21]), the same representation but with KDE detectors as proposed, and RotNet trained with an MLP head and evaluated with KDE detector as proposed. In addition, we evaluate the performance of contrastive representations without (Vanilla) and with distribution augmentation (DistAug) as proposed. For comparison, we include the results of CAVGA $R_u$ [76], which is also evaluated under the same setting. Below we highlight some points from our results:

1. The proposed two-stage framework, which trains a RotNet and constructs an one-class classifier, clearly outperforms an end-to-end framework, which trains the RotNet and uses a built-in rotation classifier [20, 21].
2. The proposed modification of RotNet with MLP projection head further improves the representation of RotNet for one-class classification.

---

[7]To be more specific, we apply transposed convolution to 57$\times$57 score map using a single convolution kernel of size 32$\times$32 whose weights are determined by Gaussian distribution and stride of 4.

3. The proposed distribution augmentation (rotation) improves the performance of contrastive representations.

4. We improve the localization AUC upon [76] both in detection and localization, both on object and texture categories.

5. Both RotNet and contrastive representations are effective for "object" categories, but not as much for "texture" categories. As reviewed by [77], semantic and texture anomaly detection problems are different and may require different tools to solve. The methods relying on geometric transformations are effective in learning representations of visual objects [46] and thus more effective for semantic anomaly detection, but less effective for texture anomaly detection.

### B.4 QUALITATIVE RESULTS ON LOCALIZATION

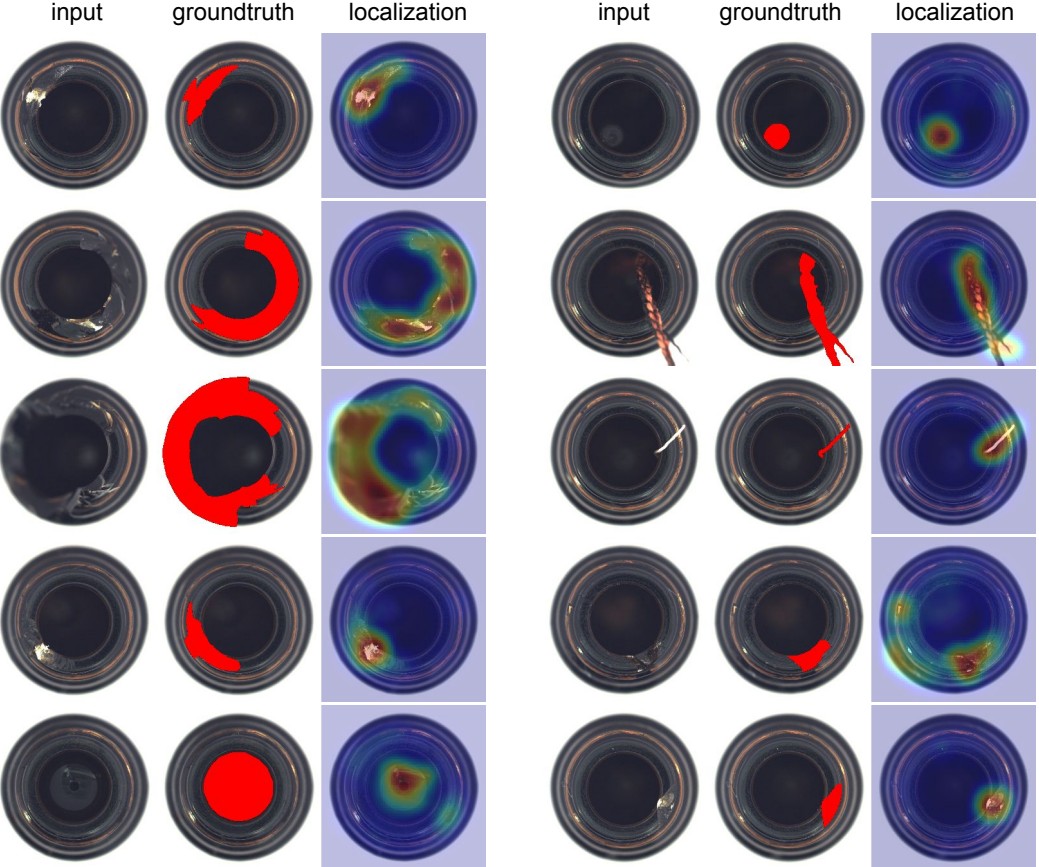

Figure 16: Visualization of defect localization on bottle category of MVTec dataset [31] using representations trained with rotation prediction on patches. From left to right, defective input data in test set, ground-truth mask, and localization visualization via heatmap.

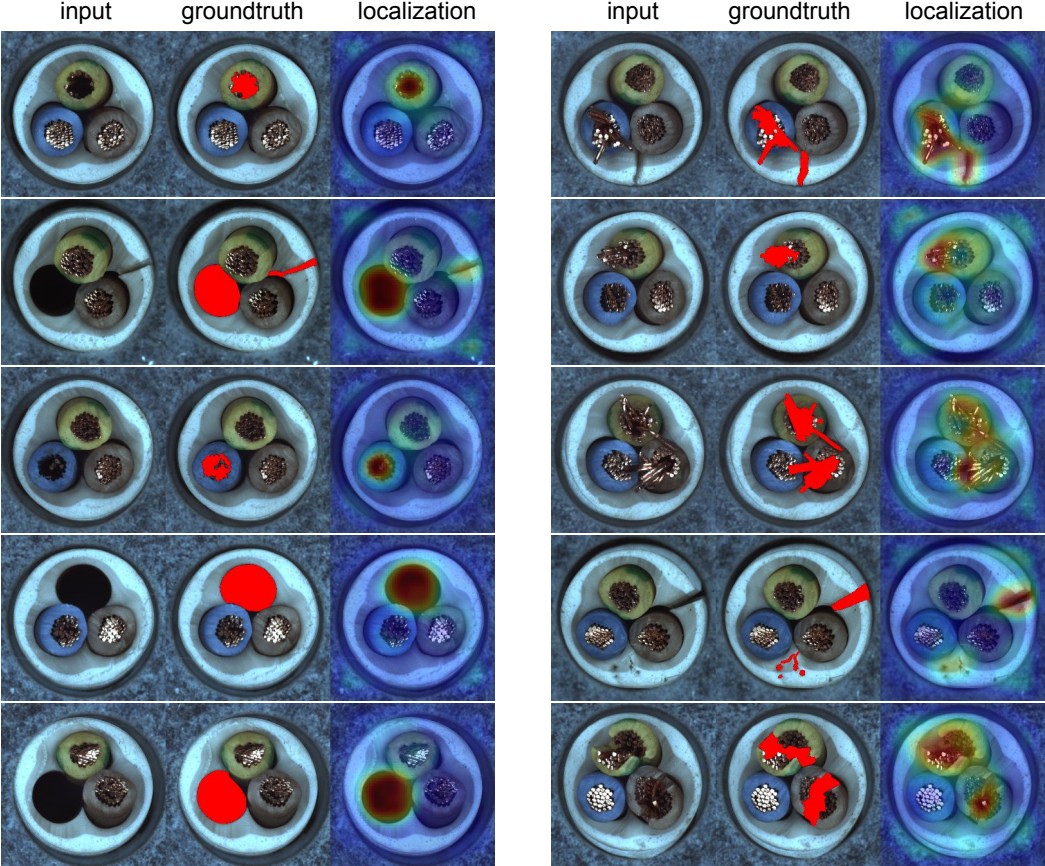

Figure 17: Visualization of defect localization on cable category of MVTec dataset [31] using representations trained with rotation prediction on patches. From left to right, defective input data in test set, ground-truth mask, and localization visualization via heatmap.

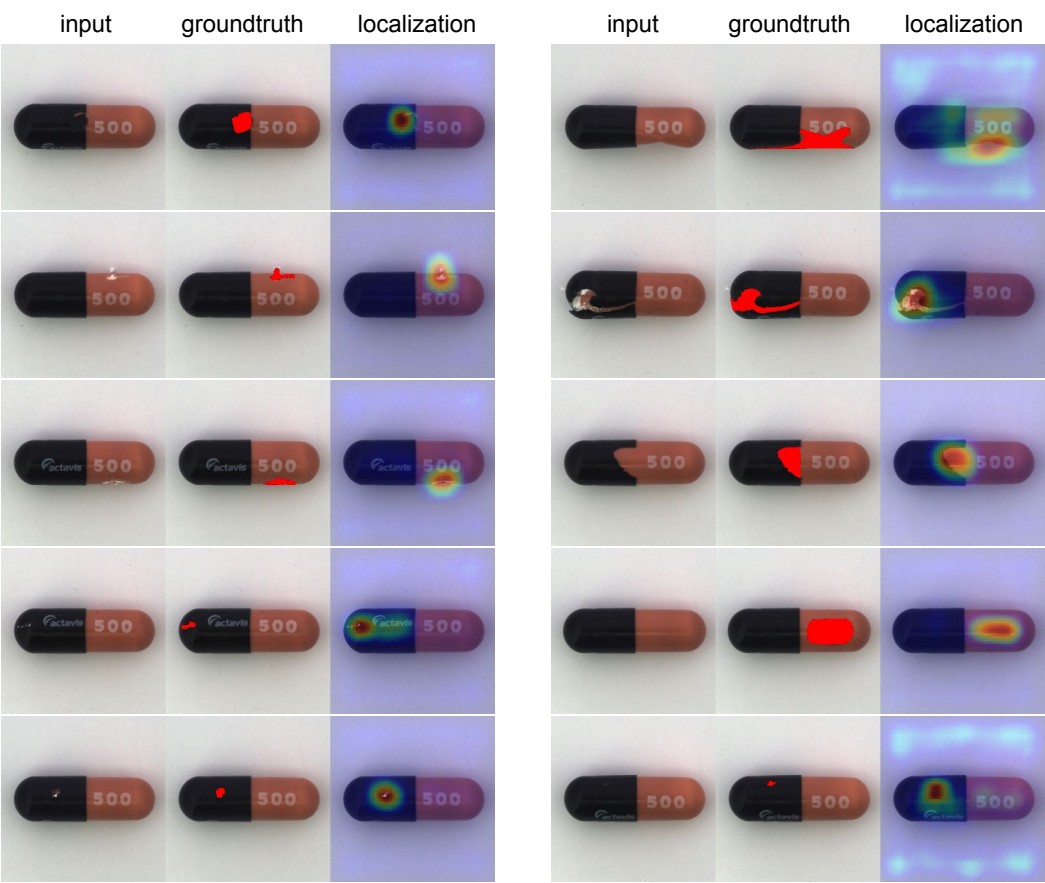

Figure 18: Visualization of defect localization on capsule category of MVTec dataset [31] using representations trained with rotation prediction on patches. From left to right, defective input data in test set, ground-truth mask, and localization visualization via heatmap.

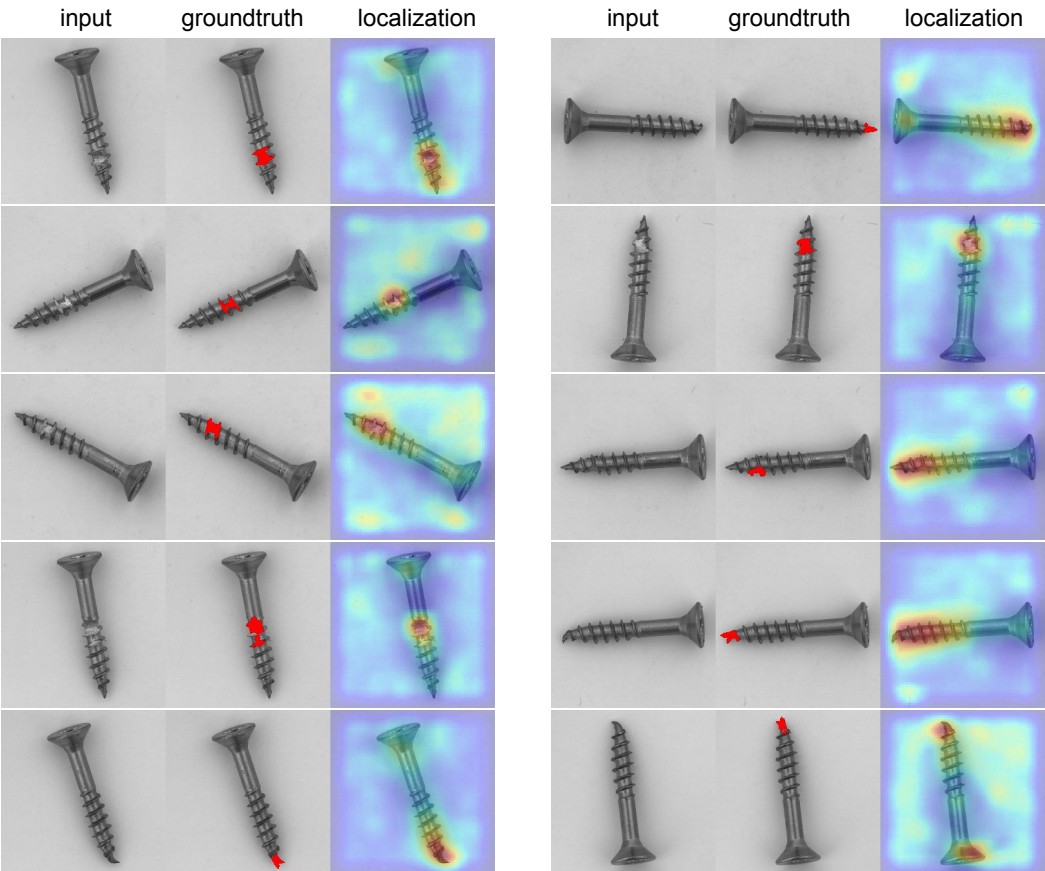

Figure 19: Visualization of defect localization on screw category of MVTec dataset [31] using representations trained with rotation prediction on patches. From left to right, defective input data in test set, ground-truth mask, and localization visualization via heatmap.

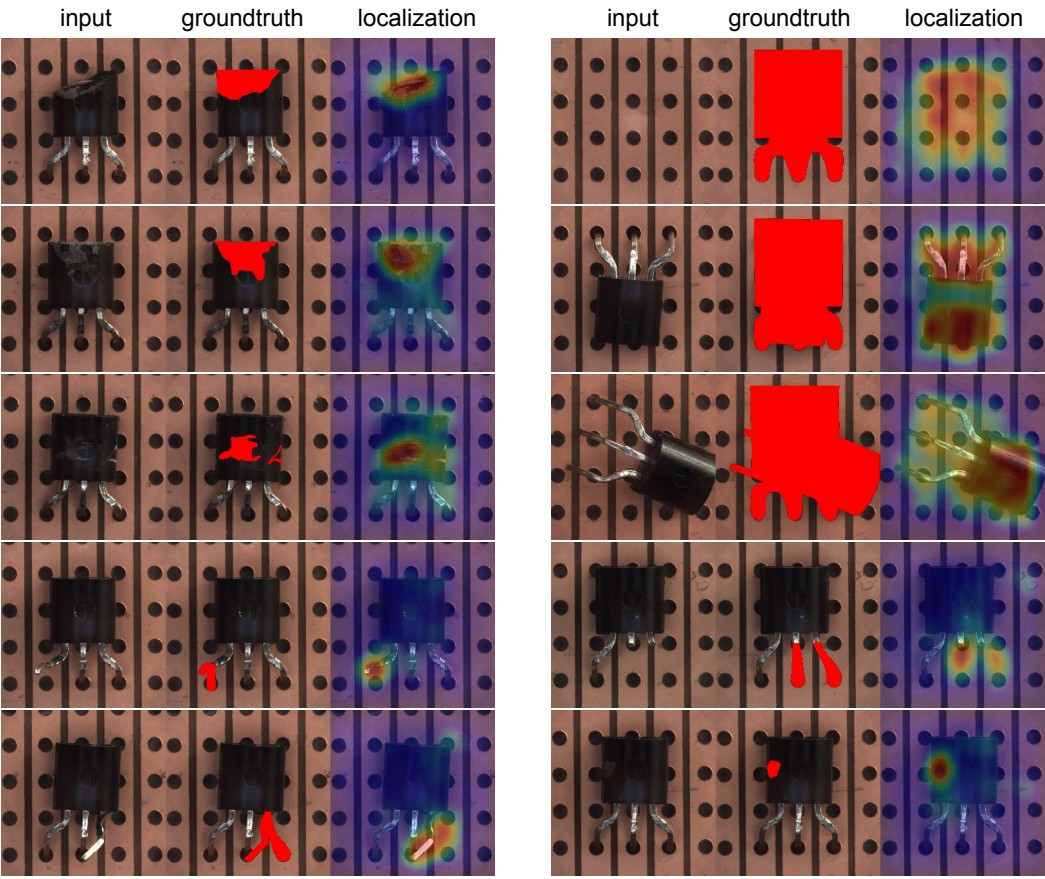

Figure 20: Visualization of defect localization on transistor category of MVTec dataset [31] using representations trained with rotation prediction on patches. From left to right, defective input data in test set, ground-truth mask, and localization visualization via heatmap.

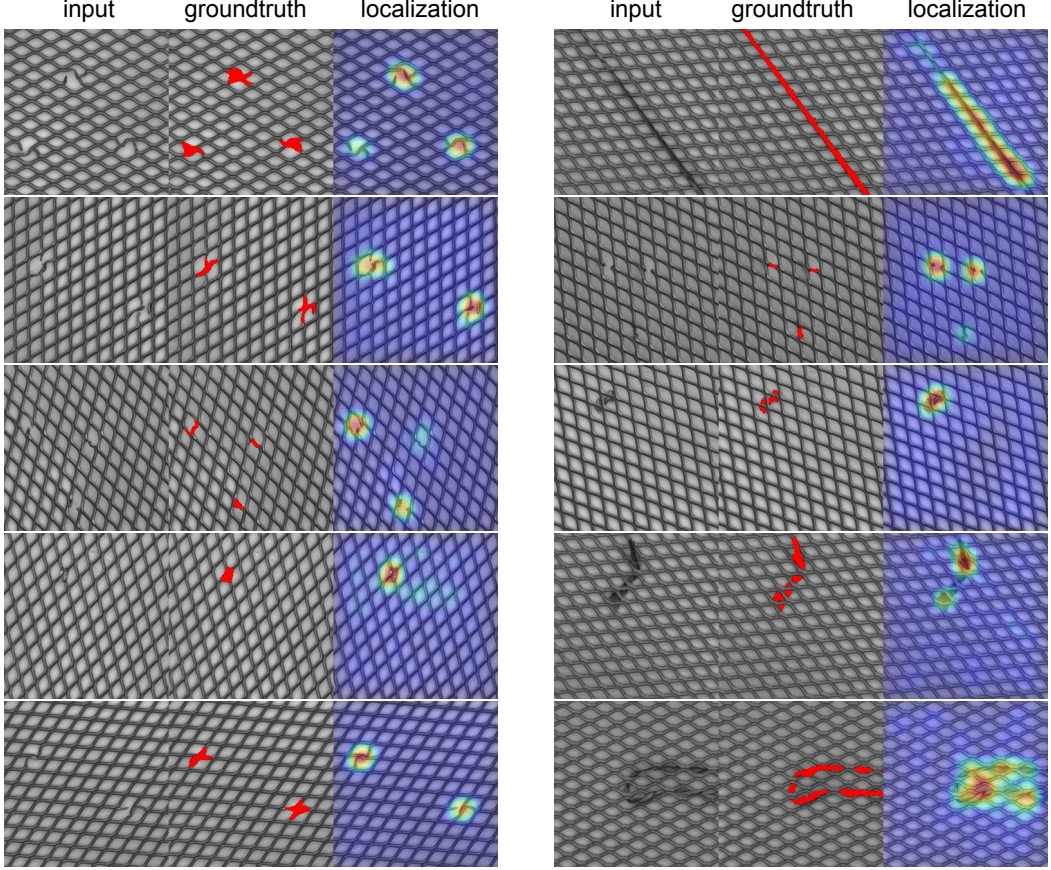

Figure 21: Visualization of defect localization on grid category of MVTec dataset [31] using representations trained with rotation prediction on patches. From left to right, defective input data in test set, ground-truth mask, and localization visualization via heatmap.

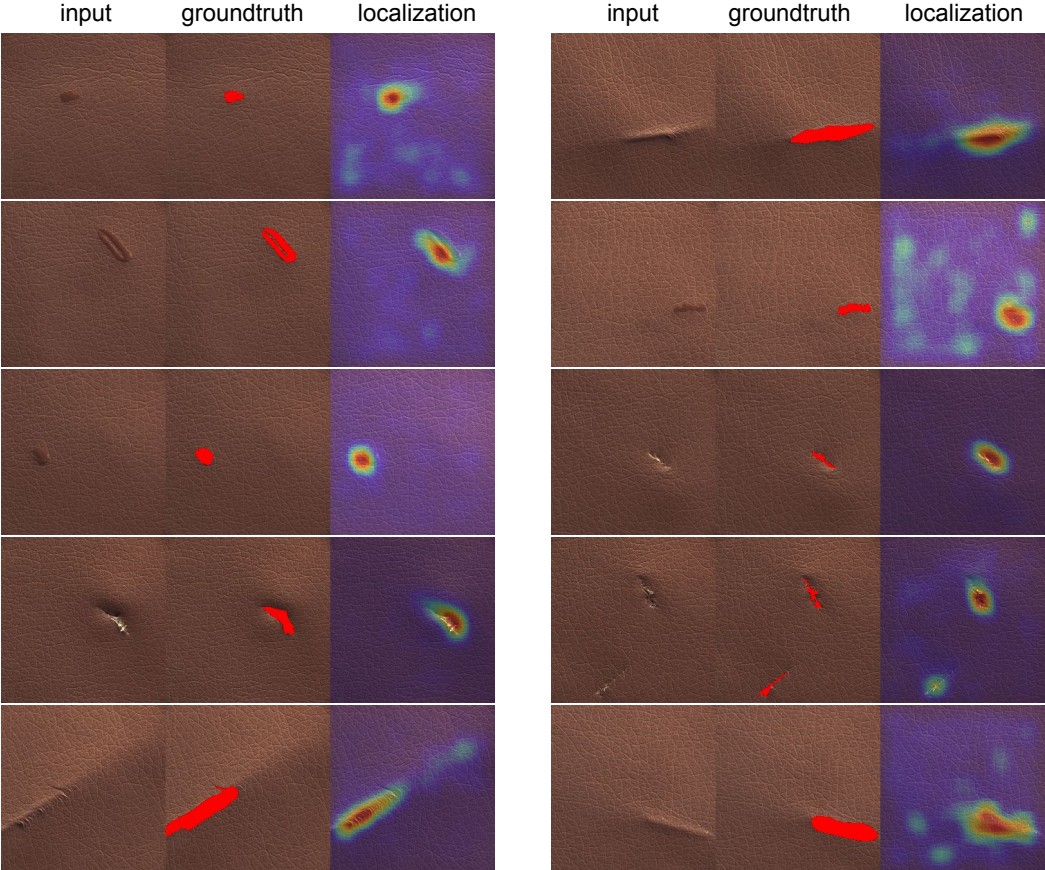

input    groundtruth    localization        input    groundtruth    localization

Figure 22: Visualization of defect localization on leather category of MVTec dataset [31] using representations trained with rotation prediction on patches. From left to right, defective input data in test set, ground-truth mask, and localization visualization via heatmap.

