# OpenReview forum: "Learning and Evaluating Representations for Deep One-Class Classification"
_ICLR.cc/2021/Conference — ICLR 2021 Poster_

### Official Review · AnonReviewer3 · 2020-10-26
**Data-augmented self-supervision with contrastive learning for anomaly detection**

**Rating:** 6
**Confidence:** 4

**Review:**

This paper proposes an anomaly detection approach that has two stages: a first stage for learning a feature representation and a second stage to train either a one-class classifier based on OC-SVM or KDE.  The main contribution of the paper is the feature representation learning that relies on contrastive learning to optimise a self-supervised loss function which minimises the distance of the samples from the same image augmented with different data augmentation functions and maximises the distance of samples from different images augmented with the same augmentation functions.  The data augmentation functions used were horizontal flip and rotation (0,90,180,270).  Results on the public datasets CIFAR-10, CIFAR-100, Fashion MNIST, and Cat-vs-Dog show that the proposed method has better anomaly detection (measured with AUC) than the state of the art.  The paper also displays qualitative anomaly detection results and an ablation study that shows: a) how close to uniform distribution (on hypersphere) the feature representations are as a function of batch size, and b) how AUC is affected with batch size and depth of MLP project heads.

This paper has outstanding results on the datasets CIFAR-10, CIFAR-100, Fashion MNIST, and Cat-vs-Dog, but it is missing result on a challenging dataset, such as Mvtec [51].  It is also missing results on anomaly localisation (e.g., Venkataramanan, Shashanka, et al. "Attention Guided Anomaly Detection and Localization in Images." arXiv preprint arXiv:1911.08616 (2019)), so it scores slightly below acceptance for results given that it is hard to assess how the method would perform in a more realistic anomaly detection problem.  In terms of the proposed method, it is quite similar to [20,21], with the difference that it uses more data augmentation functions and rely on contrastive loss.  Therefore, it scores slightly below acceptance on novelty as well.

One argument that seems contradictory is the one for class collision and uniformity.  In particular, if pre-training forces all inlier samples to be on a hyper-sphere, wouldn't it be advantageous to have a uniform distribution given that outliers could be easily detected as not lying on the hyper-sphere?  Of course, this would probably require a change in the OC-SVM classifier.  Can the authors comment on that?

Also, the argument on Sec. 2.1.3, on the effect of projection heads, says that "I(g(f(x));x) <= I(f(x);x), so f can retain more information than g, thus more suitable for downstream tasks that are not necessarily correlated with the proxy tasks".  If we push this argument, then I(f(x);x) <= I(x;x), so we should use x for downstream tasks.  Can the authors comment on that?

---

> ### Author Response · Authors · 2020-11-21
> **Clarification on the contribution, MVTec results (1/2)**
>
> We thank the reviewer for constructive feedback! Please see below our responses.
>
> > In terms of the proposed method, it is quite similar to [20,21], with the difference that it uses more data augmentation functions and rely on contrastive loss.
>
> We clarify that our contribution is not limited to proposing new representation learning methods, but proposing a two-stage framework composed of self-supervised representation learning followed by building one-class classifiers. We share the similarity with [20, 21] in learning representations when restricting ourselves to learning by predicting geometric transformations, but we construct separate one-class classifiers on learned representation. Table 2 row 2 “RotNet [20]” clearly shows the advantage of using one-class classifier over rotation classifier for one-class classification. Such a two-stage framework allows a diverse range of representation learning methods, not limited to predicting geometric transformations but also contrastive learning, to be employed to construct one-class classifiers.
>
> > missing result on a challenging dataset, such as Mvtec [51] and anomaly localisation.
>
> Thanks for your suggestion for additional evaluation on MVtec dataset. We first note that, as reviewed by [Ruff et al.](https://arxiv.org/abs/2009.11732) [A], a set of problems that our methods demonstrated effectiveness falls into a category of “semantic anomaly detection”, while some categories of MVtec data fall into a “texture anomaly detection”. They are two very different problems and may require different tools to solve. For example, while self-supervised learning by rotation prediction exploits the geometric structure of visual objects and therefore has demonstrated its effectiveness mostly on visual object recognition ([Gidaris et al.](https://arxiv.org/abs/1803.07728) [B]), texture images contain repetitive patterns without much geometric structure, so are less suitable. In this sense, we believe that evaluation on the MVTec dataset should be considered supplemental as our work is clearly focusing on semantic one-class classification.
>
> Nevertheless, we conduct experiments on MVTec dataset. Please take a look at Appendix B (from page 20) of an updated paper for description on the experimental setting and results including visualization of defect localization. From the table below, we highlight the followings of our claim:
> - The proposed two-stage framework, which trains a RotNet and constructs an one-class classifier, clearly outperforms an end-to-end framework, which trains the RotNet and uses a built-in rotation classifier [20, 21].
> - The proposed modification of RotNet with MLP projection head further improves the representation of RotNet for one-class classification.
> - The proposed distribution augmentation (rotation) improves the performance of contrastive representations.
> - Both RotNet and contrastive representations are effective for object categories, but not as much for texture categories.
> - We improve the localization AUC upon [Venkataramanan et al.](https://arxiv.org/abs/1911.08616) [C], referred by the reviewer, both in detection and localization, both on object and texture categories.
>
> | AUC                     | [CAVGA $R_u$](https://arxiv.org/abs/1911.08616) [C] | RotNet using rotation classifier [20, 21] | RotNet + KDE (proposed) | RotNet (MLP head) + KDE (proposed) | Vanilla Contrastive | Distaug Contrastive |
> |-------------------------|-----------------|-------------------------------------------|-------------------------|------------------------------------|---------------------|---------------------|
> | Detection on object     | 83.8            | 77.9                                      | 85.9                    | 89.0                               | 83.8                | 88.6                |
> | Detection on texture    | 78.2            | 73.2                                      | 75.5                    | 81.0                               | 73.0                | 82.5                |
> | Detection on all        | 81.9            | 71.0                                      | 83.5                    | 86.3                               | 80.2                | 86.5                |
> | Localization on object  | -               | 74.1                                      | 95.7                    | 96.4                               | 91.7                | 94.4                |
> | Localization on texture | -               | 78.8                                      | 86.4                    | 86.3                               | 73.4                | 82.5                |
> | Localization on all     | 89              | 75.6                                      | 92.6                    | 93.0                               | 85.6                | 90.4                |

---

> > ### Author Response · Authors · 2020-11-21
> > **Clarification on the contribution, MVTec results (2/2)**
> >
> > [A] Ruff et al., [A Unifying Review of Deep and Shallow Anomaly Detection](https://arxiv.org/abs/2009.11732), 2020.
> >
> > [B] Gidaris et al., [Unsupervised Representation Learning by Predicting Image Rotations](https://arxiv.org/abs/1803.07728), ICLR 2018.
> >
> > [C] Venkataramanan et al., [Attention Guided Anomaly Detection and Localization in Images](https://arxiv.org/abs/1911.08616), ECCV 2020.
> >
> > > One argument that seems contradictory is the one for class collision and uniformity. In particular, if pre-training forces all inlier samples to be on a hyper-sphere, wouldn't it be advantageous to have a uniform distribution given that outliers could be easily detected as not lying on the hyper-sphere? Of course, this would probably require a change in the OC-SVM classifier. Can the authors comment on that?
> >
> > We first clarify that, as in Section 2.1.3 and Figure 1, we train contrastive representations with a deep MLP projection head ($g(\cdot)$) and discard the head for OCC evaluation. When using an MLP projection head, $f(x)$ in Equation 2 should be $g = g\circ f(x)$. For OCC evaluation presented in the paper, **we always use normalized representations**, i.e., $\mathrm{Normalize}(f(x))$ or $\mathrm{Normalize}(g\circ f(x))$ (e.g., Figure 4). Therefore, there shouldn’t be a concern raised by the reviewer that the good OCC performance may have come from distinguishing the norm of representations between normal and outlier. To avoid further confusion, we will use $g(x)$ and $\mathrm{Normalize}(g(x))$ to denote unnormalized and normalized contrastive representations, instead of $f(x)$ that is a representation before the MLP projection head, in our response.
> >
> > We conducted experiments on $\mathrm{Normalize}(g(x))$ and $g(x)$ representations. Since $\mathrm{Normalize}(g(x))$ is a normalized representation, we have $g(x) = \|g(x)\|*\mathrm{Normalize}(g(x))$ and the only additional information from $g(x)$ is the norm $\|g(x)\|$. As we see below, the performance of $\mathrm{Normalize}(g(x))$ is only 63.7 AUC for a vanilla contrastive model, which is only a bit better than a chance. This suggests that, when projected to a unit hypersphere, representations between normal and outlier are not distinguishable due to uniformity. In this case, as pointed by the reviewer, the norm of an embedding plays a more important role, achieving only 70.9 AUC using $g(x)$. While improving upon using normalized representation, the performance is still unsatisfactory as it is still dominated by $\mathrm{Normalize}(g(x))$, which is poor due to uniformity, and norm can only add limited information. On the other hand, $\mathrm{Normalize}(g(x))$ of distaug contrastive representations achieves 84.3 AUC, even better than its unnormalized counterpart (81.2 using $g(x)$). Overall, our proposed approaches in Section 2.1.2. towards making $\mathrm{Normalize}(g(x))$ less uniform in “unit” hypersphere are proven to be helpful empirically as we presented in the paper.
> >
> > | CIFAR-10, seed 1 | Contrastive, MLP | DistAug, MLP |
> > |------------------|------------------|--------------|
> > | $\mathrm{Normalize}(g(x))$ | 63.7             | 84.3         |
> > | $g(x)$                   | 70.9             | 81.2         |
> > | $\mathrm{Normalize}(f(x))$  | 88.7             | 92.9         |
> >
> > > Also, the argument on Sec. 2.1.3, on the effect of projection heads, says that "I(g(f(x));x) <= I(f(x);x), so f can retain more information than g, thus more suitable for downstream tasks that are not necessarily correlated with the proxy tasks". If we push this argument, then I(f(x);x) <= I(x;x), so we should use x for downstream tasks. Can the authors comment on that?
> >
> > We clarify that our goal is to learn a good, compact high-level representation that is useful for the downstream task. The importance of high-level representations (not raw data) for downstream tasks has long been studied even before the deep learning era, such as handcrafted visual (e.g., [SIFT](https://www.cs.ubc.ca/~lowe/papers/iccv99.pdf) [A], [HOG](https://lear.inrialpes.fr/people/triggs/pubs/Dalal-cvpr05.pdf) [B]) or audio (e.g., [MFCC](http://www.haskins.yale.edu/sr/SR047/SR047_07.pdf) [C]) representations. Therefore, although we aim to preserve more information, it is under an implicit condition that it has to be a compact, high-level representation. We will clarify it in the revision.
> >
> > [A] Lowe, [Object recognition from local scale-invariant features](https://www.cs.ubc.ca/~lowe/papers/iccv99.pdf), ICCV 1999.
> >
> > [B] Dalal and Triggs, [Histograms of oriented gradients for human detection](https://lear.inrialpes.fr/people/triggs/pubs/Dalal-cvpr05.pdf), CVPR 2005.
> >
> > [C] Mermelstein, [Distance measures for speech recognition, psychological and instrumental](http://www.haskins.yale.edu/sr/SR047/SR047_07.pdf), 1976.

---

> > > ### Comment · AnonReviewer3 · 2020-11-23
> > > **Thanks for the rebuttal**
> > >
> > > The authors clarified most of my concerns in the original review, so I'd like to raise my rating from 5 to 6.  Even with the clarification about the novelty aspect, I think the paper rates low on that criterion.  The results on MVTec are somehow surprising given that other similar works didn't produce competitive results on that dataset. The last comment on the feature information is not convincing and I suggest the authors to read Tishby's work on information bottleneck in order to clarify that.

---

### Official Review · AnonReviewer1 · 2020-10-27
**state-of-the-art one class classification with representation learning and classical 1-class models**

**Rating:** 7
**Confidence:** 4

**Review:**

This paper proposes a framework for deep one-class classification (an example application being anomaly detection).  The basic idea is to combine self-supervised representation learning (eg through a proxy task such as rotation prediction or contrastive learning), with a classical approach to one-class classification, such as one-class SVM or KDE.  This is in contrast to existing methods for deep one-class classification that use simulated outliers to form a surrogate classification loss and then train end-to-end.  The paper further improves on the first stage of representation learning, by introducing modifications to contrastive learning to make it more appropriate for one-class classification.  The main insight is to introduce distribution augmentation, where geometric transformations of images, such as rotation, are treated as separate instances, to be separated from the original view.  This is motivated from the perspective of reducing uniformity of the inliers across the unit hypersphere, to allow for better separation from outliers.

Positives:
+ strong empirical results, with improved performance over existing methods for one-class classification
+ validation of two stage framework, by showing improved performance with RotNet representation with KDE detector versus RotNet end-to-end [20]
+ validation of improvements to contrastive learning for one-class classification, such as distribution augmentation, batch size selection, use of MLP project head

Minor negatives:
- I think the paper would flow a little better if the related work section was moved earlier in the paper, rather than coming only after the detailed description of the method.
- In describing distribution augmentation and contrasting it with standard data augmentation for contrastive learning, it is clarified that the two sets of augmentations are disjoint.  I would it have found it helpful if the paper was explicit about which data augmentations were used for the contrastive learning, as this did not seem to be stated in the paper.

Overall I found this to be a nice paper with strong empirical results.

---

> ### Author Response · Authors · 2020-11-21
> **Clarification on data augmentation**
>
> We thank the reviewer for constructive feedback! Please see below our responses.
>
> > I would have found it helpful if the paper was explicit about which data augmentations were used for the contrastive learning, as this did not seem to be stated in the paper.
>
> As mentioned in Section 2.1., we follow [Chen et al.](https://arxiv.org/abs/2002.05709) [A] to define data augmentation $\mathcal{A}$: resize and crop, horizontal flip, color jittering, gray-scale and gaussian blur are applied in sequence. While we will clarify the reviewer's concern in the paper, we make [our code available](https://anonymous.4open.science/r/be40ded0-200f-41ec-808e-114c4b8b3761/) for additional detail and reproducible research.
>
> [A] Chen et al., [A simple framework for contrastive learning of visual representations](https://arxiv.org/abs/2002.05709), ICML 2020.

---

### Official Review · AnonReviewer4 · 2020-10-27
**Review of "Learning and Evaluating Representations for Deep One-Class Classification"**

**Rating:** 7
**Confidence:** 3

**Review:**

This paper presents a two-stage representation learning approach to deep one-class classification.

In the first stage, a mapping f to a versatile high-level latent representation is learned using self-supervised learning for a contrastive learning proxy task. In the second stage, the same mapping f is used to map the data to the latent space, whereafter a traditional one-class classifier such as OC-SVM or KDE, is applied.
It is shown that the one-class task puts somewhat different requirements on the representation than with a multi-class classification task, both 1) in terms of uniformity of the data points in the representation, which is desired for multi-class tasks but not fully beneficial for one-class tasks, and 2) in terms of minimizing or maximizing the distance between different instances of the negative class - for multi-class tasks you want the distances maximized, while for one-class tasks you want the negative (inlier) examples close together. 1) is addressed by using smaller batch sizes in training while 2) is addressed by distribution augmentation that will render a compact inlier distribution in the representation.

This paper is overall a good paper that will be interesting to a certain audience at ICLR.
+ It is well written, well motivated, with a clear argument and as far as I can see, technically correct.
+ The experiments are well designed, valid and exhaustive, with comparison to a range of baselines as well as an ablation study.
+ Moreover, the visual explanation of what the different representations have focused on is highly interesting.
+ I appreciate the comprehensive grounding of the contribution in both new and old related work. The reference list contains all the relevant state of the art, as well as references to more classical work such as [13,14,29,47,53].

The paper is not highly seminal, but more incremental in nature, putting together and modifying existing methodology. However, since it is very well done, the work is absolutely worth acceptance.

A criticism is that there are some repetition in the line of argument, for example between 2.1.2 second paragraph and 2.1.3 first paragraph. A more compact, e.g., section 2.1 would render more space for results which now have been pushed to the appendix to a large degree. Another suggestion for improvement could be to indicate more clearly in figure 1(b) that f is kept fixed in this step. This could be done e.g. with a different color of the f box in figure 1(b).

---

> ### Author Response · Authors · 2020-11-21
> **Thank you!**
>
> We thank the reviewer for constructive feedback! We will reflect your suggestions in the final revision.

---

### Official Review · AnonReviewer2 · 2020-10-27
**Official Blind Review**

**Rating:** 5
**Confidence:** 5

**Review:**

Summary:
They investigated the effectiveness of self-supervised learning (SSL) for one class classification (OCC).
Here is what I think are contributions relative to existing literature -
Empirically improved AUC for multiple OCC datasets, here are the techniques that were useful -
Used “distribution augmentation” [DistAug] for learning representation for OCC, and in ablation studies show DistAug leads to  improvement over standard augmentation
Used KDE and OCSVM on top of learned representation, and showed improvement over using the classification head training during SSL
Used a smaller batch size (=32)
Used a MLP head during SSL
The authors also included a section of visualizing explanation using existing techniques to illustrate how their method leads to more reasonable decisions.

Strength:
The paper is well written. I appreciate the clarity, and good coverage of the current literature.
The ablation studies are thorough, which make the empirical improvement solid.

Concerns:
The uniformity argument is weak. The authors state the empirical improvement on OCC using their method hinges on the DistAug technique, which is motivated to reduce the uniformity of the learned representation. When achieved the inliers will live in the dense regions on the hypersphere, and outliers will live on the non-occupied region.  This assumes all the test inputs are projected onto the hypersphere, including the outlier.  From my understanding, the authors used f() for OCC, not \phi() which is the normalized (i.e. hypersphere) output. In this case, there are many ways that OCC can be achieved even if \phi() of the training inputs are uniform on the hypersphere.  Suppose  both the inliers and outliers after f() live on hyperspheres, just with a different radius, then  after normalization they can both be uniformly distributed  on  the  same hypersphere.
One question is if there is a difference in using f()  or \phi() for OCC.
Furthermore, the authors try to back this claim up using Figure 4, but I cannot  seem to connect the dots here.
They authors used MMD to a uniform distribution to measure how uniform the representations are.  The less uniform (i.e. higher MMD), the better it should be for OCC.  The  correlation between MMD and AUC does not  seem  to be very strong.  E.g., for the (DA) gf variant, the 2 metrics actually  seem negatively correlated.
This again, makes me wonder if “less uniformity” really is why their technique led to an improvement in OCC.
If this is not why, then we should find another explanation for why there was  an improvement.
There is always the concern that the improvement comes from extra hyperparameter tuning. Did the author also tune for good hyperparameters for the non DistAug version as described in A.3?


Overall, a fairly thorough empirical investigation into better techniques for using SSL for OCC.  It can be a decent contribution along the lines  of one of the  “improve techniques …”  papers if the above concerns can be addressed.  In fact, I think not focusing on selling DistAug, but really identifying what contributes to  the gain empirically makes this  paper stronger.


References:
[DistAug] Heewoo Jun, Rewon Child, Mark Chen, John Schulman, Aditya Ramesh, Alec Radford, and Ilya Sutskever. Distribution augmentation for generative modeling. In Proceedings ofMachine Learning and Systems 2020, pages 10563–10576, 2020.

---

> ### Author Response · Authors · 2020-11-21
> **Response to uniformity argument, hyperparameter tuning for baselines (1/2)**
>
> We thank the reviewer for constructive feedback! Please see below our responses.
>
> > The uniformity argument is weak. The authors state the empirical improvement on OCC using their method hinges on the DistAug technique, which is motivated to reduce the uniformity of the learned representation. When achieved the inliers will live in the dense regions on the hypersphere, and outliers will live on the non-occupied region. This assumes all the test inputs are projected onto the hypersphere, including the outlier. From my understanding, the authors used f() for OCC, not \phi() which is the normalized (i.e. hypersphere) output. In this case, there are many ways that OCC can be achieved even if \phi() of the training inputs are uniform on the hypersphere. Suppose both the inliers and outliers after f() live on hyperspheres, just with a different radius, then after normalization they can both be uniformly distributed on the same hypersphere. One question is if there is a difference in using f() or \phi() for OCC.
>
> We first clarify that, as in Section 2.1.3 and Figure 1, we train contrastive representations with a deep MLP projection head ($g(\cdot)$) and discard the head for OCC evaluation. When using an MLP projection head, $f(x)$ in Equation 2 should be $g = g\circ f(x)$. For OCC evaluation presented in the paper, **we always use normalized representations**, i.e., $\mathrm{Normalize}(f(x))$ or $\mathrm{Normalize}(g\circ f(x))$ (e.g., Figure 4). Therefore, there shouldn’t be a concern raised by the reviewer that the good OCC performance may have come from distinguishing the norm of representations between normal and outlier. To avoid further confusion, we will use $g(x)$ and $\mathrm{Normalize}(g(x))$ to denote unnormalized and normalized contrastive representations, instead of $f(x)$ that is a representation before the MLP projection head, in our response.
>
> We conducted experiments on $\mathrm{Normalize}(g(x))$ and $g(x)$ representations. Since $\mathrm{Normalize}(g(x))$ is a normalized representation, we have $g(x) = \|g(x)\|*\mathrm{Normalize}(g(x))$ and the only additional information from $g(x)$ is the norm $\|g(x)\|$. As we see below, the performance of $\mathrm{Normalize}(g(x))$ is only 63.7 AUC for a vanilla contrastive model, which is only a bit better than a chance. This suggests that, when projected to a unit hypersphere, representations between normal and outlier are not distinguishable due to uniformity. In this case, as pointed by the reviewer, the norm of an embedding plays a more important role, achieving only 70.9 AUC using $g(x)$. While improving upon using normalized representation, the performance is still unsatisfactory as it is still dominated by $\mathrm{Normalize}(g(x))$, which is poor due to uniformity, and norm can only add limited information. On the other hand, $\mathrm{Normalize}(g(x))$ of distaug contrastive representations achieves 84.3 AUC, even better than its unnormalized counterpart (81.2 using $g(x)$). Overall, our proposed approaches in Section 2.1.2. towards making $\mathrm{Normalize}(g(x))$ less uniform in “unit” hypersphere are proven to be helpful empirically as we presented in the paper.
>
> | CIFAR-10, seed 1 | Contrastive, MLP | DistAug, MLP |
> |------------------|------------------|--------------|
> | $\mathrm{Normalize}(g(x))$ | 63.7             | 84.3         |
> | $g(x)$                   | 70.9             | 81.2         |
> | $\mathrm{Normalize}(f(x))$  | 88.7             | 92.9         |

---

> > ### Author Response · Authors · 2020-11-21
> > **Response to uniformity argument, hyperparameter tuning for baselines (2/2)**
> >
> >
> > > Furthermore, the authors try to back this claim up using Figure 4, but I cannot seem to connect the dots here. They authors used MMD to a uniform distribution to measure how uniform the representations are. The less uniform (i.e. higher MMD), the better it should be for OCC. The correlation between MMD and AUC does not seem to be very strong. E.g., for the (DA) gf variant, the 2 metrics actually seem negatively correlated.
> > This again, makes me wonder if “less uniformity” really is why their technique led to an improvement in OCC. If this is not why, then we should find another explanation for why there was an improvement.
> >
> > We clarify that our argument is **not** “the less uniform, the better it should be for OCC” (a clear counter-example is a hypersphere-collapsed network, which maps all data points into a single representation in the hypersphere). Rather, our aim is to show the importance of *proper* batch size and DistAug training, each of which we propose to resolve the issue of nearly-perfect uniformity of contrastive representations when trained following the usual practice -- large batch training without DistAug.
> >
> > Figure 4 should be interpreted with care. First, between Contrastive (DA) $g\circ f$ (cyan) and Contrastive $g\circ f$ (green), we observe a correlation between MMD and AUC from Figure 4a and Figure 4b. We have explained this in Section 5.1 **Distribution Augmentation** paragraph. Second, taking an example of Contrastive (DA) $g\circ f$ (cyan) with different batch sizes (see Table below), we observe a correlation between MMD and AUC from Figure 4a and Figure 4b for batch size $\geq$ 32. As explained in Section 5.1 **Batch size** paragraph, when batch size $\leq$ 16, even though representations are less uniform (as shown in Figure 4a), they are less informative and less representative of the data. Our study shows that there is a fundamental trade-off between the information quantity and the uniformity of representations and parameters, such as batch size, should be properly tuned and the proposed DistAug effectively resolves the uniformity issue. Although our intuition and several empirical evidence suggests such trade-off between uniformity and information quantity for OCC with contrastive representatioins, we agree with the reviewer that there could be other reasons contributing to the performance gain. Uniformity could be one of those that has not yet studied in existing works and we believe this is an interesting future direction.
> >
> > | batch size | 32    | 64    | 128   | 256   | 512   |
> > |------------|-------|-------|-------|-------|-------|
> > | MMD        | 0.108 | 0.060 | 0.032 | 0.025 | 0.031 |
> > | AUC        | 81.6  | 78.2  | 75.5  | 75.0  | 74.9  |
> >
> > > There is always the concern that the improvement comes from extra hyperparameter tuning. Did the author also tune for good hyperparameters for the non DistAug version as described in A.3?
> >
> > We assure the reviewer that we tune the hyperparameters for the non-DistAug version as described in A.3, and transfer the discovered hyperparameters from the non-DistAug version to the DistAug version without further tuning. We make our [code](https://anonymous.4open.science/r/be40ded0-200f-41ec-808e-114c4b8b3761/) available for transparency and reproducibility.

---

> > > ### Comment · AnonReviewer2 · 2020-11-23
> > > **Thanks for the response**
> > >
> > > Thanks for the response.
> > > The clarifications about notations and details are helpful.
> > >
> > > The response about Figure 4 should be improved.  It was clear that for the lines using $f$, MMD correlates with AUC.  My point was for the 2 lines using $g$ MMD does not correlate with AUC.  This to me suggests that MMD is not a good indicator of AUC in general.
> > > Or like said in the response, potentially you should combine MMD with an "informative" measure so it becomes a good indicator of AUC.  I understand if you want to say this is left for future work, but in that case you should still be precise in the discussion of your current results.
> > >
> > > At the moment, my main concern remains.  The argument of reduced uniformity leads to improved OCC accuracy is not well supported in the paper (and response).

---

> > > > ### Author Response · Authors · 2020-11-25
> > > > **Correlation between uniformity and OCC accuracy**
> > > >
> > > > We thank the reviewer for their time and efforts!
> > > >
> > > > For clarity, our framework involves two representations, $f$ and $g = g\circ f$ as follows:
> > > >
> > > > [ input ========> $f$ ======> $g$ => contrastive loss ]
> > > >
> > > > and $\mathrm{Normalize}(f)$ is used to obtain the final OCC accuracy. Here, we also need to clarify that our uniformity analysis only applies to $g$, an embedding that is used to compute the contrastive loss, but not to $f$ since there exists a deep non-linear MLP projection head between $f$ and $g$ that complicates their relation and analysis. To reflect this point and mitigate the confusion, we removed the red and blue curves from Figure 4(a) and revised the text accordingly.
> > > >
> > > > To show the correlation between the uniformity of $g$ and OCC accuracy, we draw scatter plots in Figure 7 (page 15) of an updated manuscript, where x-axis is log(MMD) and y-axis is AUC on CIFAR-10. The data points in the scatter plots are obtained from 5 runs using different random seeds of vanilla and distAug contrastive representations with batch sizes in [32, 64, 128, 256, 512]. We observe a strong positive correlation (i.e., the lower the MMD, the more uniformly distributed, and therefore the lower the AUC) with Pearson’s correlation coefficient of **0.914** between the log(MMD) and the AUC, where both metrics are evaluated using $\mathrm{Normalize}(g)$. We also plot and measure the correlation between log(MMD) evaluated using $\mathrm{Normalize}(g)$ and AUC evaluated using $\mathrm{Normalize}(f)$. In addition, we also observe highly positive correlation between the MMD using $g$ and the AUC using $f$ (**0.774** Pearson’s correlation coefficient), suggesting that reducing uniformity of $g$ improves the OCC accuracy tested with both $g$ and $f$ representations.

---

### Author Response · Authors · 2020-11-21
**Author response**

We thank the positive recognition from the reviewers on the well-written draft and a comprehensive empirical analysis. In addition to the proposed distribution augmentation contrastive learning, we highlight our contribution in proposing a two-stage framework for one-class classification, which is flexible to use different representation learning algorithms. The [manuscript](https://openreview.net/pdf?id=HCSgyPUfeDj) is updated by adding results on MVTec datasets as requested by Reviewer 3, where we outperformed existing works pointed by Reviewer 3. We also include the [link to the anonymized code repo](https://anonymous.4open.science/r/be40ded0-200f-41ec-808e-114c4b8b3761/). The code will be made publicly available upon acceptance. We reply to each reviewer below the review.

---

### Comment · ~nima_rafiee1 · 2021-09-08
**Is the claim for the SOTA results correct? Ablation study on OC-SVM and KDE**

According to the paper, the authors claim that they achieved SOTA results for one-class classification. But compared to the paper CSI: Novelty Detection via Contrastive Learning on Distributionally Shifted Instances- which is also referred to by the authors- the results are not SOTA.
e.x for CIFAR-10 and CIFAR-100 the results achieved by the authors are 92.5 and 86.5 which are less than 94.3 and 89.6 reported in the CSI paper.  I wonder if there is any reason that authors refer to this paper in the text but avoid comparing their results in table 2?

I also wonder if you have any ablation study on removing OC-SVM and KDE and use only cosine similarity as the score function on the learned representations.

---

### Decision · Program_Chairs · 2021-01-07
**Final Decision**

**Decision:**

Accept (Poster)

**Comment:**

This paper investigates the one-class classification problem, proposing to learn a self-supervised representation and a distribution-augmented contrastive learning method; thorough results and analysis show that the method is effective and backs up their claims in terms of the underlying mechanism for why it works. In general, reviewers thought the paper was well-written, well-motivated/argued, and presents a thorough related work comparison and experimentation, though the novelty was found to be somewhat low. Several reviewers brought up some possible weaknesses in terms of demonstrating uniformity of the representations as well as suggesting additional datasets. Through an interesting discussion, the authors provided additional visualizations and results on the Mvtec dataset. This further bolstered the arguments in the paper.

Overall, this is a strong paper with a clear argument and contribution, and so I recommend acceptance.